# KITTEN 🐱: A KNOWLEDGE-INTENSIVE EVALUATION OF IMAGE GENERATION ON VISUAL ENTITIES

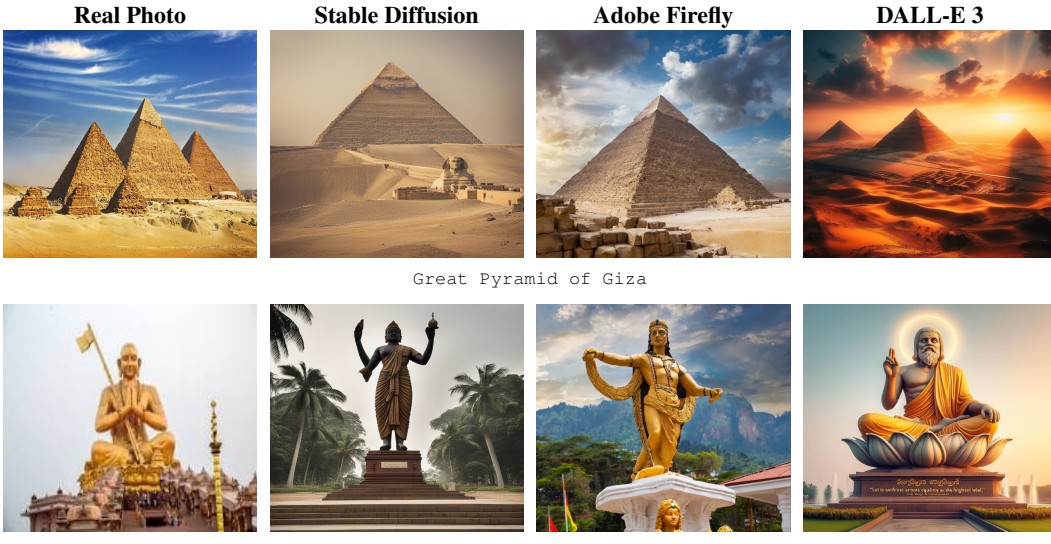

Figure 1: **Can text-to-image models generate precise visual details of real-world entities?** State-of-the-art image generation models are effective at rendering well-known entities (*i.e.*, Great Pyramid of Giza), but often struggle with less-known entities and end up with hallucinated depictions.

## ABSTRACT

Recent advancements in text-to-image generation have significantly enhanced the quality of synthesized images. Despite this progress, evaluations predominantly focus on aesthetic appeal or alignment with text prompts. Consequently, there is limited understanding of whether these models can accurately represent a wide variety of realistic visual entities — a task requiring real-world knowledge. To address this gap, we propose a benchmark focused on evaluating **K**nowledge-**I**n**T**ensive image genera**T**ion on real-world **EN**tities (*i.e.*, KITTEN). Using KITTEN, we conduct a systematic study on the fidelity of entities in text-to-image generation models, focusing on their ability to generate a wide range of real-world visual entities, such as landmark buildings, aircraft, plants, and animals. We evaluate the latest text-to-image models and retrieval-augmented customization models using both automatic metrics and carefully-designed human evaluations, with an emphasis on the fidelity of entities in the generated images. Our findings reveal that even the most advanced text-to-image models often fail to generate entities with accurate visual details. Although retrieval-augmented models can enhance the fidelity of entity by incorporating reference images during testing, they often over-rely on these references and struggle to produce novel configurations of the entity as requested in creative text prompts.

## 1 INTRODUCTION

Recent advancements in generative AI have revolutionized multimedia content creation. Large Language Models (LLMs) have demonstrated impressive capabilities in assisting with knowledge-

intensive tasks such as question-answering and summarization. Cutting-edge image generation models, such as Imagen (Saharia et al., 2022; Imagen 3 Team, 2024; Hu et al., 2024), DALL-E (Ramesh et al., 2021; 2022) and Stable Diffusion (Rombach et al., 2022), can produce photorealistic and creative images from textual descriptions. However, as these generative models become more capable and popular, it is crucial to assess their reliability. Recent research has focused on the factuality of LLMs, revealing that even the most advanced models can generate errors or inaccurate content, potentially undermining trust and causing societal harm (Muhlgay et al., 2023; Feng et al., 2023).

Despite the growing attention to factuality issues in large language models, significantly less focus has been placed on the factual accuracy of image generation models. Existing benchmarks for text-to-image generation predominantly assess alignment with general textual descriptions (Lin et al., 2015), compliance with image-editing instructions (Ku et al., 2024), or adherence to specified spatial relationships (Gokhale et al., 2022). These benchmarks fall short in evaluating how well models can generate images that accurately reproduce the precise visual details of real-world entities, objects, and scenes grounded in trustworthy knowledge sources (see examples in Figure 1).

Recently, HEIM (Lee et al., 2024) introduces an evaluation suite that examines various aspects of image generation models, including their ability to generate entities such as historical figures or well-known subjects. However, the real-world distribution of visual entities is far richer than the entities evaluated by HEIM, and a more breadth assessment is demanded. In addition, the evaluation of HEIM primarily measures the alignment between the generated images and the text prompts associated with the names of entities, which often can not capture the fine-grained visual details essential for assessing the reproduction of visual-world knowledge. Since many nuances of real-world entities cannot be conveyed through text alone, directly evaluating the fidelity of entities in the generated images is crucial.

To fill this critical gap in evaluating whether the image generation models can reproduce visual world knowledge, we introduce KITTEN, a benchmark dataset and evaluation suite specifically designed to assess how well these models can generate visually accurate representations of real-world entities grounded in trust-worthy knowledge sources. Unlike prior benchmarks that primarily focus on aesthetics, general text alignment, or commonsense reasoning, KITTEN leverages prompts derived from visual entities documented in Wikipedia (Hu et al., 2023a), a widely recognized and trustworthy knowledge base, and evaluates real-world entities over 8 visual domains (*i.e.*, aircraft, vehicle, cuisine, flower, insect, landmark, plant, and sport in Figure 2). This ensures that the generated images are evaluated based on comparison with reliable and verifiable visual information crowdsourced by the large internet population. Moreover, we have developed a comprehensive set of human evaluation criteria that emphasize the precise and accurate visual depiction of these entities, capturing subtle yet critical details essential for visual correctness. By directly assessing the fidelity of entities in the generated images against established knowledge, KITTEN aims to promote the evaluation of world knowledge for image generation models.

Using KITTEN, we conduct a comprehensive evaluation of various text-to-image models, including both standard models and customization models fine-tuned or employing in-context learning with retrieved reference images (Chen et al., 2022). Our assessment focuses on the fidelity of entities in the generated images, particularly in depicting real-world entities across eight visual domains. The results reveal that even the most advanced text-to-image models (Imagen 3 Team, 2024; Black Forest Labs, 2024) frequently fail to produce correct representations, often generating images with inaccuracies or missing critical details essential for correctness in visual knowledge. While retrieval-augmented models show promise in enhancing precise visual details by incorporating reference images during testing, they tend to over-rely on these references. This dependency hampers their ability to create novel configurations of entities as requested in creative text prompts. These findings highlight a significant challenge in current image generation models: balancing the fidelity of entity with creative flexibility. Our study underscores the need for advanced techniques that can generate precise visual details in images without compromising the ability to respond to diverse and imaginative user inputs.

## 2 FIDELITY OF ENTITIES IN TEXT-TO-IMAGE GENERATION

We provide an overview of existing text-to-image evaluation benchmarks, and show that the fidelity of entities in generated images is often overlooked in existing evaluation. Next, we discuss the motivation and design considerations for evaluating the fidelity of entities in text-to-image generation.

**Existing evaluation for text-to-image generation.** The evaluation of text-to-image generation models has been a longstanding challenge, and numerous efforts have been made to more accurately measure model performance. Fréchet Inception Distance (FID) (Heusel et al., 2017) is one of the most commonly used metrics to evaluate the perceptual quality of model outputs by measuring the distribution gap between generated images and real-world images. For text-image alignment, CLIP-T scores (Hessel et al., 2021) are widely used, evaluating the CLIP feature similarity between the generated image and the input prompt. These metrics typically provide a summary of the quality of the generated images. Several works evaluate the alignment between generated images and their text descriptions (Yarom et al., 2024; Gordon et al., 2023; Hu et al., 2023b; Wiles et al., 2024; Cho et al., 2023). However, these efforts primarily focus on general semantic consistency rather than fine-grained visual accuracy of depicted entities and specialized visual-world knowledge.

Recent works aim to more thoroughly evaluate models by decomposing the evaluation into multiple sub-categories, such as attribute binding and numeracy, and propose corresponding benchmarks for these categories. For example, Gokhale et al. (2022) propose the SR$_{2D}$ dataset and VISOR metric to benchmark spatial relationships in text-to-image models, measuring whether the objects in the generated image adhere to the specified spatial relationship (*e.g.*, an orange *above* a giraffe). Huang et al. (2023) introduce T2I-CompBench++, which consists of 8,000 compositional text prompts from four categories (*e.g.*, attribute binding) and eight sub-categories (*e.g.*, color binding), along with corresponding metrics. Bakr et al. (2023) propose HRS-Bench, which evaluates text-to-image model performance across 13 skills categorized into five major groups (*i.e.*, bias, fairness, generalization, accuracy, and robustness). Hu et al. (2023b) introduce TIFA v1.0, a benchmark with 4,000 text inputs and 25,000 questions across 12 categories, paired with an automatic evaluation metric that measures image faithfulness via visual question answering. Additionally, Ku et al. (2024) proposes ImagenHub, which evaluates models across different generation tasks by measuring the semantic consistency and perceptual quality of the generated images.

**Fidelity of entities in text-to-image generation.** While text-to-image models enable the possibility of generating creative images from text descriptions, this can become problematic when visual-world knowledge is required, i.e., generating the correct visual details of entities. Existing works have identified this issue and proposed solutions to mitigate hallucination (Lim & Shim, 2024). However, there is still no clear methodology to systematically assess the limitations of these models, which is crucial for improving this aspect. In this work, we propose KITTEN, a benchmark consisting of a diverse range of subjects across various real-world entities such as landmarks, plants, and animals. Using KITTEN, we conduct a systematic study on the fidelity of entities in text-to-image generation models, focusing on their ability to accurately depict a wide range of visual entities. We evaluate the latest text-to-image models and retrieval-augmented customization models using both automatic metrics and carefully designed human evaluations, emphasizing the fidelity of entities in the generated images.

## 3 KITTEN BENCHMARK

To understand the reliability of text-to-image models in generating knowledge-intensive concepts, we design the KITTEN benchmark to evaluate the fidelity of entities in image generation models.

### 3.1 DESIGN DESIDERATA OF KITTEN

The key to creating the benchmark is constructing a set of image-generation prompts that require grounding in visual-world knowledge. There are two specific properties that differentiate our benchmark from prior evaluation frameworks of image generation:

- Specialized knowledge: while many existing evaluation frameworks aim to test the common-sense knowledge of image generation models such as spatial or physical relationships (Gokhale et al., 2022; Huang et al., 2023), we would like to stress-test the image generation models by focusing on generating items from specific domains. Therefore, we create the benchmark using image concepts from Wikipedia, a rich source of knowledge-intensive data source. Specifically, we use the OVEN-Wiki (Hu et al., 2023a) dataset, which contains several domain-specific entities and their corresponding images such as landmarks, plants, and animals.
- Fidelity of entity: while most existing image generation benchmarks focus on evaluating the instruction-following capability of the models, we would like to understand how well these mod-

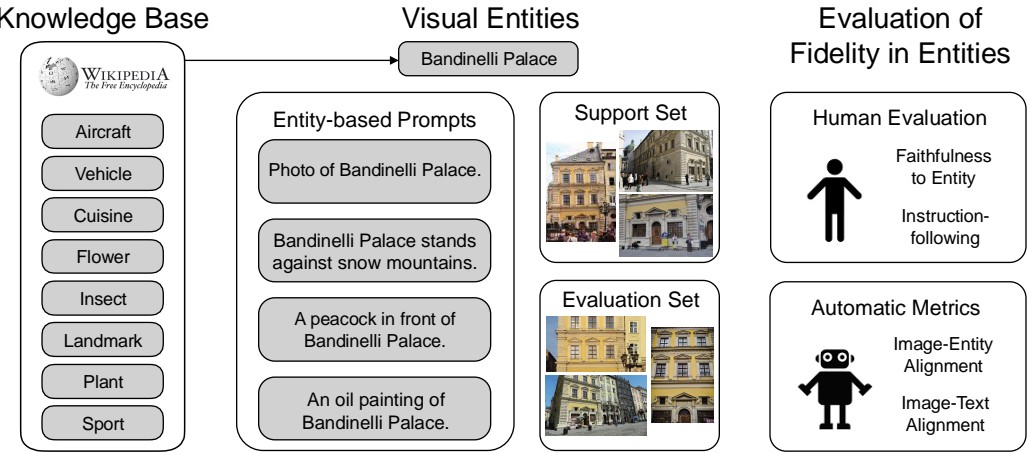

Figure 2: **KITTEN benchmark** is constructed from real-world entities over eight domains. For each selected entity, we construct four types of image-generation prompts based on the entity. KITTEN includes a support set of entity images from the knowledge source for evaluating retrieval-augmented models. It also contains an evaluation set for assessing the fidelity of the generated entities.

els are at faithfully representing real-world concepts grounded in visual knowledge sources such as Wikipedia. Therefore, we design a specific set of human evaluations targeted at capturing the visual fidelity of generated entities grounded on specialized knowledge sources.

Guided by the above principles, next, we clarify the details of the KITTEN benchmark.

## 3.2 CREATING ENTITY-BASED PROMPTS

To create a diverse set of prompts focusing on faithfulness to knowledge-grounded concepts, we select several categories of entities from the OVEN-Wiki (Hu et al., 2023a) dataset as the basis for the image generation prompts. Figure 2 shows the process of creating the benchmark. We select entities from eight specialized domains, including human-made objects (aircraft, vehicle, landmark, cuisine), natural species (flower, plant, insect), and human activities (sport).

After selecting the entities for the benchmark, we construct four types of prompts evaluating the models' ability to respond to diverse and imaginative user inputs.

- The basic prompt: `Photo of Bandinelli Palace.`
- The entity in a specified place: `Bandinelli Palace stands against snow mountains.`
- The composition of the entity: `A peacock in front of Bandinelli Palace.`
- The entity in specific styles / materials: `An oil painting of Bandinelli Palace.`

For each entity in the selected categories, we collect a set of reference images from Wikipedia for human evaluation. We also provide a support set of entity images, so that we can evaluate retrieval-augmented models that can take advantage of external knowledge sources (see discussions on evaluated models in Section 4). Detailed statistics of the data can be found in Section A.5.

## 3.3 HUMAN EVALUATION

We design a set of human evaluations targeting the visual fidelity of the target entity in the generated image. We propose to decompose the evaluation into two different aspects: 1) faithfulness to the prompt entity; and 2) adherence to prompt instructions beyond the prompt entity. This design allows the human raters to focus on two clearly defined criteria. It also enables more informative analysis and comparison of different models, as our results show that there are often trade-offs between these two aspects. We show the reference images of the prompt entity but also encourage human raters to conduct their own research to verify the faithfulness of the entity. We use the average of 5 raters for each generated image as the final human evaluation score. The human annotation interface is shown in Figure 3. We also include open-ended questions to gather qualitative feedback. Full rater instructions are shown in Section A.6.

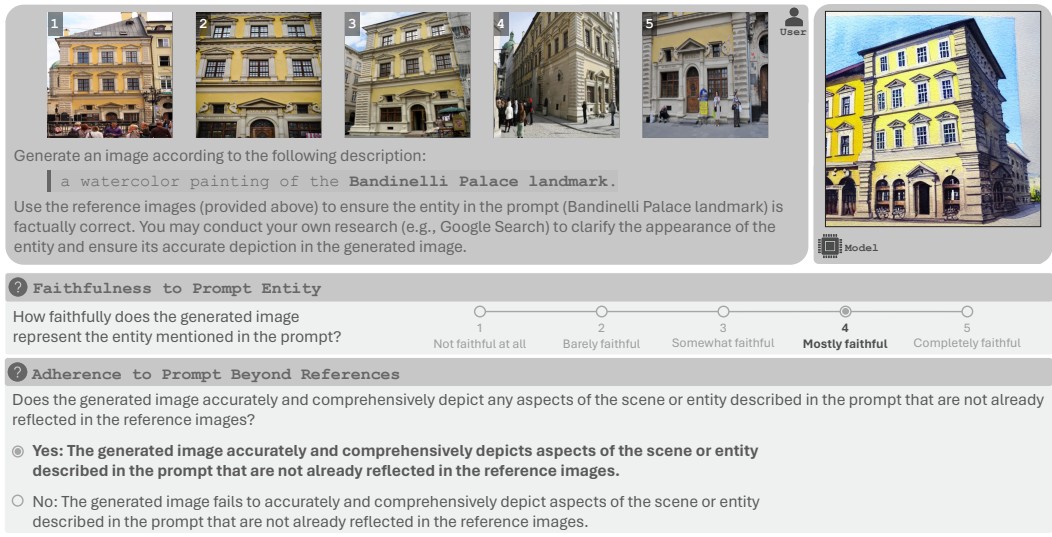

Figure 3: **Annotation interface.** Raters are asked to: 1) rate the image's faithfulness to the prompt entity on a 1-5 scale, and 2) indicate whether the image follows the prompt with a yes or no response.

- **Faithfulness to Entity.** For the evaluation of faithfulness to the prompt entity, we use a scale of 1 to 5 where 5 is completely faithful to the prompt entity and 1 means the generated image has no similarity at all to the prompt entity.

- **Instruction-following.** We simply use a yes/no question to evaluate whether the generated image adheres to the prompt instructions beyond the entity. We calculate the percentage of prompts with an answer of yes.

## 3.4 AUTOMATIC METRICS

We also gather the results of popular automatic metrics for image generation models which mostly measure the similarity of the generated images with the references or prompts. While metrics are not specialized to capture the visual fidelity of the generated entity, we still include their results to provide a comprehensive analysis of whether these widely used metrics are suitable to evaluate the fidelity of entities in the generated images.

- **Image-Text Alignment**. We use the *CLIP-T Score* (Hessel et al., 2021) to measure the alignment between the generated image and the text prompt. This score is the cosine similarity between the image and text features in CLIP's feature space (Radford et al., 2021).

- **Image-Entity Alignment**. To assess the faithfulness between the entities in the generated image and reality, we measure the average pairwise visual similarity between the generated image and reference images of the target entity in the evaluation set, as a proxy for how closely fine-grained details match. We compute the cosine similarity in the self-supervised DINO's feature space (Oquab et al., 2024).

## 4 EVALUATED MODELS

We would like to provide a comprehensive set of analysis using KITTEN to understand the visual-world knowledge learned in current state-of-the-art models. Here we provide details of the text-to-image models evaluated in the paper.

## 4.1 TEXT-TO-IMAGE BACKBONE MODELS

First, we examine general text-to-image backbone models that directly generate images solely based on text prompts without using additional tools or reference images.

- Stable Diffusion (Rombach et al., 2022) contains an autoencoder that maps images to a latent space, and then a diffusion model is trained in this latent space to reduce computational demands.

- Imagen (Saharia et al., 2022) uses a T5-based text encoder and cascade conditional diffusion models to generate high-resolution images, showing that advanced language understanding improves image fidelity and text-image alignment.
- Imagen-3 (Imagen 3 Team, 2024) is a successor to Imagen, known for handling long and descriptive prompts and generating rich image details.
- Flux (Black Forest Labs, 2024) is a successor to Stable Diffusion, incorporating multimodal and parallel diffusion transformer blocks with flow matching training, enhancing the ability to handle complex compositions.

## 4.2 RETRIEVAL-AUGMENTED TEXT-TO-IMAGE MODELS

The retrieval-augmented method is a family of image generation approach that leverages some support images (*e.g.*, retrieved by the search engine) to augment the model (via either fine-tuning or in-context learning), and improve the fidelity of entities in its generation. Our goal is to evaluate this family of models and identify whether methods that incorporate those *support* reference images into its modeling could enhance the fine visual fidelity of the entity generation. Particularly, we study a few models listed below:

- DreamBooth (Ruiz et al., 2023) **fine-tunes** the Stable Diffusion model to learn a special token encoded the target entity, using an objective of reconstructing the reference images. It then generates the entity in new contexts using prompts that include the special token.
- Custom-Diff (Kumari et al., 2023), similar to DreamBooth, **fine-tunes** partial weights of the backbone Stable Diffusion model to achieve entity-focused generation.
- Instruct-Imagen (Hu et al., 2024) is an image generation model that follows multi-modal instructions. We use instruct-Imagen for entity customization through **in-context learning**, by encoding reference entity images into instructions: `Generate an image of <entity_name>, referring to the images <ref_image_1>, ..., <ref_image_K>, and follow the caption: <prompt>.`

Specifically, we provide some ground-truth reference entity images (hold-out from the entity images used for evaluation) as the support images to augment the above methods, and then sample new images from them following the evaluation text prompt.

## 5 EVALUATION RESULTS

In this section, we present the evaluation results and analysis of various text-to-image models. We begin with a discussion of the human evaluation results, followed by an examination of popular automatic metrics, and explore how they correspond with the human evaluations.

### 5.1 HUMAN EVALUATION

**Improvement in faithfulness and reduction in instruction-following.** Figure 4 (Top) shows that retrieval-augmented models—Custom-Diff, DreamBooth, and Instruct-Imagen—generally exhibit much better faithfulness to the entity compared to their base models, SD and Imagen. This is because these models can incorporate reference images during testing, allowing them to generate visual concepts that are not well-represented in the base models' parameters. However, we find that retrieval-augmented models tend to have reduced instruction-following capabilities compared to their backbone models.

Although the pattern of higher faithfulness and lower instruction-following is consistent across retrieval-augmented methods, the extent of the impact varies significantly depending on the specific method used. In particular, Instruct-Imagen shows a marked increase in faithfulness score (2.81→4.22) but also a substantial reduction in instruction-following score (72.2→46.5). This suggests that while retrieval-augmented methods enhance the model's ability to generate images that are more faithful to the entities by incorporating reference images during testing, they often over-rely on these references and struggle to create novel configurations of the entity as requested in creative text prompts.

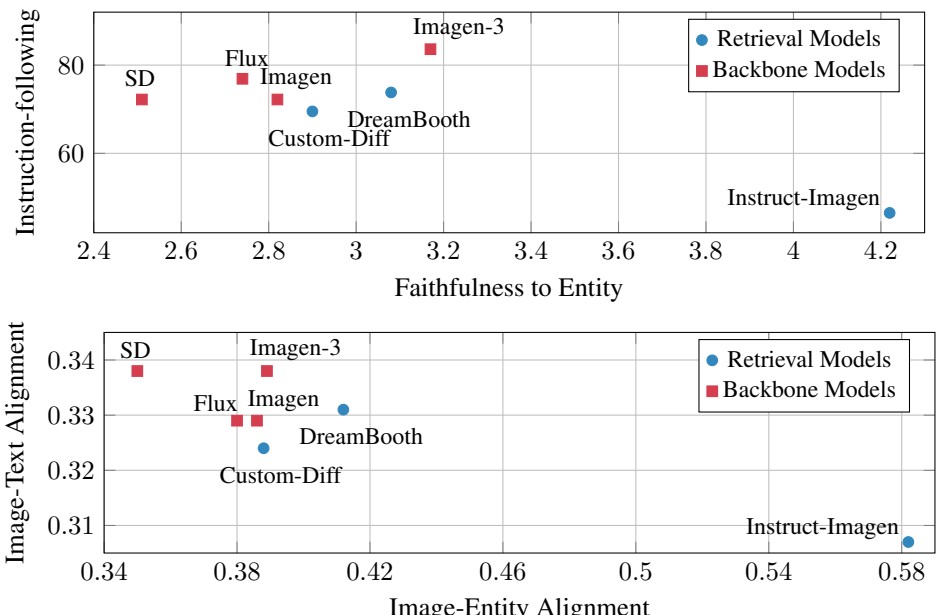

Figure 4: **Evaluation results** of different text-to-image models. Top: Human evaluation showing the trade-off between Faithfulness and Instruction-following. Bottom: Automatic metrics showing the relationship between Image-Entity and Image-Text Alignments.

**High faithfulness with strong backbone models.** Imagen-3 achieves the highest instruction-following score (83.6) and a high faithfulness score (3.17), outperforming even the retrieval-augmented models DreamBooth (3.08) and Custom-Diff (2.90). This demonstrates that Imagen-3, a strong backbone model, is capable of generating specialized entities based solely on text prompts, without relying on reference images. However, there still exists a significant gap in the fidelity of entities compared to the highest faithfulness score obtained by Instruct-Imagen (4.22). These findings highlight the need for advanced techniques that improve the fidelity of entities without compromising the model's ability to respond to diverse user inputs.

**Improved faithfulness without harming instruction-following.** The retrieval model DreamBooth improves entity faithfulness compared to its baseline model, SD (2.51→3.08), without compromising SD's instruction-following score (72.2→73.8), shifting its performance to the right in the figure. Applying DreamBooth to the advanced backbone model Imagen-3 could further enhance faithfulness, potentially achieving a sweet spot that balances the fidelity of entities with creative flexibility.

**Impact of backbone model improvements.** The results indicate that improvements to base models alone can enhance both instruction-following and faithfulness. For example, Flux outperforms its predecessor SD by 0.23, and Imagen-3 surpasses its predecessor Imagen by 0.35 in faithfulness. However, these faithfulness improvements are still minor compared to those achieved by retrieval-augmented methods, such as DreamBooth, which shows a 0.57 improvement over SD. This suggests that simply enhancing the text-to-image model is not sufficient; additional methods like retrieval augmentation are necessary to achieve higher levels of faithfulness. Interestingly, we observe that the Imagen family, Imagen (2.82) and Imagen-3 (3.17), consistently achieve higher faithfulness scores compared to the SD family, SD (2.51) and Flux (2.74). Detailed human evaluation scores across domains can be found in Section A.2.

## 5.2 AUTOMATIC METRIC

**Improvement in image-entity alignment and reduction in image-text alignment.** Figure 4 (Bottom) shows that retrieval models increase the image-entity alignment score while decreasing the image-text alignment compared to their respective base models. In particular, Instruct-Imagen stands out with a substantial increase in the image-entity score (0.386→0.582) and the most significant drop

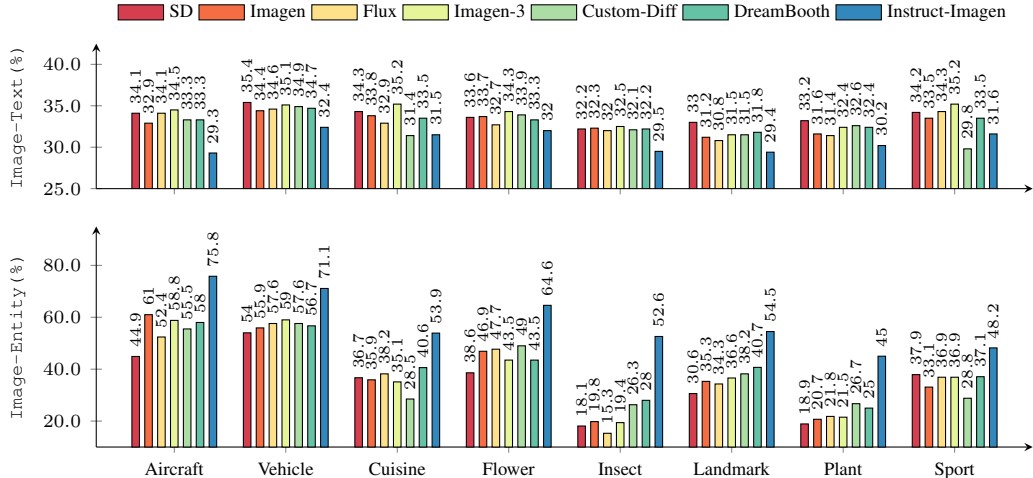

Figure 5: **Performance across domains.** The model performance depends on the domain-specific content. Retrieval models perform better in the insect, landmark, and plant domains and get worse in cuisine and sport domains in the image-entity score, which might be due to the entities' occurrence in common image datasets.

in the image-text score (0.329→0.307). These observations align with the human evaluation, where retrieval-augmented models show improved faithfulness in entity but reduced instruction-following. In addition, this trend is consistent with observations in recent works (Materzyńska et al., 2023), which indicate that models incorporating additional inputs, such as reference images, tend to have lower image-text alignment scores compared to base models. This is due to a trade-off between aligning with the text and with the images. With longer training times, alignment to reference images improves, while alignment to the text decreases.

**Subtle variations with backbone model improvements.** In contrast to the findings from human evaluations, we observe that improving base models does not necessarily lead to gains in the automatic metrics. For example, Flux (0.329) performs worse than its predecessor SD (0.338) in the image-text metric, and Imagen-3 (0.389) shows only a marginal improvement over Imagen (0.386) in the image-entity score. These findings suggest that automatic metrics have a limited ability to capture meaningful variations between models of similar quality, even when architectural enhancements or training improvements are made.

**Misalignment between automatic metrics and human evaluation.** While the overall observations from the automatic metrics align with the human evaluation, there are notable discrepancies. For instance, although DreamBooth achieves a higher instruction-following score in the human evaluation compared to its base model SD, it has a lower image-text alignment score. We hypothesize that the image-text score may not accurately assess the alignment between the generated image and rare entities. For example, with the prompt "The Teufelsmauer landmark shimmers in the sunlight," it is unclear whether the image-text similarity for "Teufelsmauer" is evaluated correctly. This highlights a potential gap in the traditional metrics (Lee et al., 2024), where it fails to measure true alignment between the unique entity and the generated image. On the other hand, Imagen-3 ranks higher in faithfulness in the human evaluation, yet DreamBooth outperforms Imagen-3 in DINO scores, showing a misalignment between human perception and the learned semantic features.

**Performance across domains.** Figure 5 shows that the performance of each method is domain-dependent. Retrieval-augmented models generally achieve higher image-entity alignment scores than backbone models in the insect, landmark, and plant domains. These domains contain less frequent terms in common image datasets (e.g., LAION), meaning that these visual concepts are not well-represented in the base model's parameters. The retrieval-augmented models perform better by incorporating reference images during inference. In contrast, the retrieval-augmented method, Custom-Diff, performs worse than its base model, SD, in the cuisine and sport domains. These domains contain common terms, such as snowboarding and guacamole, which the SD model has well-memorized. The Custom-Diff model's performance degrades, potentially due to fine-tuning

on a smaller reference set. This variability suggests that the effectiveness of a retrieval-augmented method may be influenced by the nature of the domain-specific content, as well as the model and customization method used. The results indicate that the optimal choice of retrieval-augmented method remains an open question and can be task or domain-specific. Detailed automatic metrics across domains can be found in Section A.3.

## 5.3 ABLATION STUDY

**Selection of image-entity alignment metrics.** We conduct an ablation study to select the most appropriate image-entity alignment metric in Table 1. Two popular visual features are tested for calculating cosine similarity scores between reference and generated images: *CLIP-Image* (Radford et al., 2021) and *DINO* (Oquab et al., 2024).

We find that DINO scores provide a clearer separation between models compared to CLIP-Image scores, making it a more discriminative metric for assessing image-entity alignment. For example, the difference between Custom-Diff and Instruct-Imagen is much larger when using DINO (0.19) compared to CLIP-Image (0.11), suggesting that DINO is better at capturing subtle differences in faithfulness. This may be due to DINO's focus on primary entities, allowing for a more accurate estimation of similarity between generated images and reference content.

Table 1: Image-entity metrics.

| Model | CLIP-I | DINO |
|---|---|---|
| SD | 0.646 | 0.350 |
| Imagen | 0.646 | 0.386 |
| Flux | 0.639 | 0.380 |
| Imagen-3 | 0.650 | 0.389 |
| Custom-Diff | 0.643 | 0.388 |
| DreamBooth | 0.674 | 0.412 |
| Instruct-Imagen | 0.751 | 0.582 |

**Correlation between automatic metrics and human evaluation.** To quantify the consistency between automatic metrics and human evaluation, we computed Pearson and Spearman correlations in Table 2. CLIP-Text and CLIP-Image show moderate alignment with user evaluations.

Compared to CLIP-I metrics, DINO demonstrates stronger alignment with user evaluations of faithfulness, with increasing Pearson ($0.239 \rightarrow 0.510$) and Spearman ($0.340 \rightarrow 0.504$) correlations. This suggests that DINO is more effective at measuring faithfulness than CLIP-Image. However, the variability in correlations varies across categories, such as the negative correlation between CLIP-I and faithfulness in the plant domain, highlighting that more precise evaluation metrics are still needed to accurately reflect human judgments. Detailed correlation scores across domains can be found in Section A.4.

Table 2: Correlation between automatic metrics and human evaluation.

| Automatic | Human | Type | Average |
|---|---|---|---|
| CLIP-T | Instruction | Pearson | 0.337 |
| | | Spearman | 0.384 |
| CLIP-I | Faithfulness | Pearson | 0.239 |
| | | Spearman | 0.340 |
| DINO | Faithfulness | Pearson | 0.510 |
| | | Spearman | 0.504 |

## 5.4 QUALITATIVE RESULTS

We present the visual results in Figure 6. In the above example, the backbone models—SD, Flux, and Imagen—exhibit lower faithfulness to the entity, with mismatches in the tail fin, engine, and wing shapes compared to the reference image. In contrast, retrieval-augmented models like Custom-Diff, DreamBooth, and Instruct-Imagen, which use support images during testing, show better visual alignment with the target. Surprisingly, Imagen-3, despite relying only on text prompts, generates accurate visual details due to its strong language understanding capability. These observations align with the evaluation results in Figure 4.

In the example below, Custom-Diff and DreamBooth, show reduced instruction-following compared to their backbone model, SD. In particular, Custom-Diff struggles to create novel compositions like placing the target entity "next to a giant sandcastle." In contrast, the backbone models excel in instruction-following and entity faithfulness, likely because "Rolls-Royce Phantom Drophead Coupé" is well-represented in their training data. The retrieval-augmented models underperform due to over-relying on the reference images and knowledge forgetting during fine-tuning on small support sets. These results suggest that the success of retrieval-augmented methods highly depends on the entity domain and customization approach. More results can be found in Section A.1.

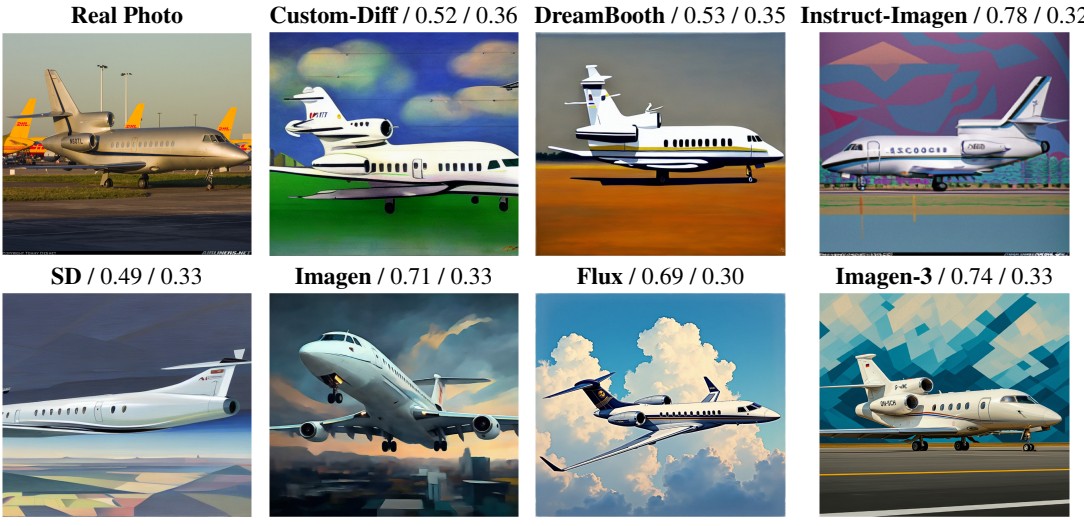

*A painting of Dassault Falcon 900 in a cubist style, parked in a surreal landscape.*

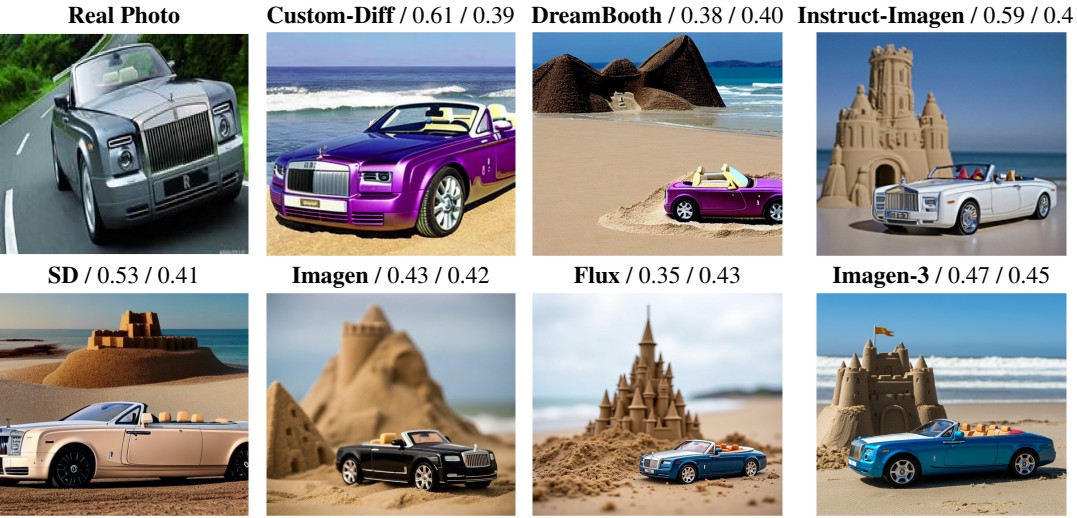

*A toy Rolls-Royce Phantom Drophead Coupé next to a giant sandcastle on the beach.*

Figure 6: **Qualitative results.** Top: The backbone models show lower faithfulness to the entity, with mismatches in the tail fin, engine, and wing shapes. Bottom: The retrieval models struggle with instruction-following, over-relying on references and failing to create novel compositions. We show the DINO and CLIP-T scores together with the generated results.

## 6 CONCLUSION

In this paper, we propose the KITTEN— a benchmark for evaluating the fidelity of entities in text-to-image generation focusing on visual concepts that require specialized knowledge. We construct carefully-designed image generation prompts focusing on domain-specific entities grounded in Wikipedia, and we design a visual faithfulness human evaluation framework by targeting the fidelity of entities in the generated images. Extensive analysis is conducted on several varieties of state-of-the-art image generation models using the proposed framework. Our results indicate that while advanced backbone models can generate specialized entities, there remains a notable gap in the fidelity of entities compared to retrieval-augmented models. Although retrieval-augmented methods enhance faithfulness to entities, they often struggle with creative prompts. This underscores the need for techniques that improve the fidelity of entities without sacrificing the ability to follow instructions.

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

# A APPENDIX

In this appendix, we first present additional qualitative results from the KITTEN benchmark for both the backbone and retrieval-augmented models. Second, we provide detailed human evaluation scores, including metrics for entity faithfulness and instruction-following across eight domains. Third, we offer a comprehensive breakdown of automatic metric scores for image-text alignment and image-entity alignment. Fourth, we include the per-domain correlation scores between human evaluations and automatic metrics. Fifth, we present statistics for our KITTEN benchmark. Finally, we provide the instructions given to human raters for evaluating the generated images in the KITTEN benchmark.

## A.1 ADDITIONAL QUALITATIVE EXAMPLES

We present additional qualitative examples evaluating the backbone models—SD, Imagen, Flux, and Imagen-3—as well as the retrieval models—Custom-Diff, DreamBooth, and Instruct-Imagen—across various domains in the KITTEN benchmark. The results for the aircraft, vehicle, cuisine, flower, insect, landmark, plant, and sport domains are shown in Figure 7, Figure 8, Figure 9, Figure 10, Figure 11, Figure 12, Figure 13, Figure 14, respectively.

Figure 9 presents the results for the cuisine domain. For the prompt, "A Wakame dish with a cherry flower on top of it," retrieval-based models such as Custom-Diff, DreamBooth, and Instruct-Imagen successfully generate the entity Wakame. However, Custom-Diff fails to capture the composition of the cherry flower. In contrast, the backbone models—Imagen, Flux, and Imagen-3—correctly generate the cherry flower, but the representation of Wakame does not align with its real-world appearance. In the example prompt, "photo of a Hot and sour soup," the backbone models introduce incorrect ingredients that significantly deviate from a real-world hot and sour soup, demonstrating how these models hallucinate entities instead of reproducing real-world knowledge accurately.

Figure 11 shows the results for the insect domain. For the prompt, "Satyrium liparops sitting at the beach with a view of the sea," all models correctly generate the beach scene as instructed. However, the backbone models hallucinate the insect's appearance, generating a completely different insect, while the retrieval models accurately depict the visual details of the Satyrium liparops. For the prompt, "Promachus hinei wearing sunglasses," the backbone models demonstrate strong instruction-following ability, with Imagen and Imagen-3 correctly composing the sunglasses on the insect. In contrast, the retrieval models fail to generate this composition. However, the backbone models do not generate an accurate representation of the insect itself, while the retrieval models correctly generate the entity.

Figure 13 presents the results for the plant domain. In the example, "An impressionistic painting of Cirsium andersonii," none of the models effectively follow the prompt in generating the "impressionistic painting," highlighting the challenge of rendering specific artistic styles. However, the retrieval models generate the entity more accurately than the backbone models. In another example, "Penstemon rydbergii on a rustic wooden table next to a rose plant," although the generated entity lacks the fine details of the reference, the backbone models successfully capture the composition of "next to a rose," demonstrating their stronger instruction-following ability.

These observations align with the quantitative results, further demonstrating that while advanced backbone models can generate specialized entities, there is a notable gap in the fidelity of entities compared to retrieval-augmented models. Although retrieval-augmented methods improve faithfulness to entities, they often struggle with creative prompts, highlighting the need for techniques that enhance entity accuracy without compromising instruction-following abilities.

## A.2 DETAILS OF THE HUMAN EVALUATION SCORES

In Figure 4 (Top), we present the human evaluation results for backbone and retrieval-augmented text-to-image models, comparing their performance in terms of entity faithfulness and instruction-following accuracy. We provide a detailed breakdown of the human evaluation results across eight domains of the KITTEN benchmark in Table 3 and Table 4.

We observe that the performance of each method varies across domains. For instance, while the overall faithfulness scores of Custom-Diff and DreamBooth are lower than the backbone model Imagen-3, these retrieval models consistently perform better in the insect, landmark, and plant do-

mains. When comparing backbone models, Flux outperforms SD in the overall score; however, SD demonstrates higher faithfulness in the insect, landmark, and plant domains, suggesting that Flux struggles to generate less frequent entities. Additionally, the retrieval models, DreamBooth and Custom-Diff, improve faithfulness over SD in some cases but show reduced performance in the cuisine and sport domains. In terms of instruction-following score, Instruct-Imagen performs notably lower in the flower (31.3) and insect (21.9) domains, while Imagen-3 achieves significantly higher scores in the cuisine (93.8) and sport (96.9) domains.

Table 3: Detailed breakdown of human evaluation: faithfulness to entity.

| Model | Aircraft | Vehicle | Cuisine | Flower | Insect | Landmark | Plant | Sport | Overall |
|---|---|---|---|---|---|---|---|---|---|
| ■ SD | 2.07 | 3.57 | 3.42 | 2.36 | 1.61 | 1.95 | 2.09 | 3.02 | 2.51 |
| ■ Imagen | 3.19 | 3.62 | 3.55 | 3.21 | 1.54 | 2.11 | 2.04 | 3.3 | 2.82 |
| ■ Flux | 3.51 | 3.89 | 3.35 | 2.52 | 1.44 | 1.77 | 1.83 | 3.59 | 2.74 |
| ■ Imagen-3 | 3.74 | 4.04 | 4.26 | 3.37 | 1.63 | 2.31 | 1.98 | 4.04 | 3.17 |
| ● Custom-Diff | 2.59 | 3.90 | 2.52 | 3.82 | 2.37 | 3.02 | 2.85 | 2.10 | 2.90 |
| ● DreamBooth | 3.26 | 3.88 | 3.23 | 3.18 | 2.46 | 3.06 | 2.64 | 2.93 | 3.08 |
| ● Instruct-Imagen | 4.23 | 4.70 | 4.43 | 4.48 | 3.78 | 4.06 | 4.02 | 4.08 | **4.22** |

(■: Text-to-Image Models, ●: Retrieval-augmented Models)

Table 4: Detailed breakdown of human evaluation: instruction-following.

| Model | Aircraft | Vehicle | Cuisine | Flower | Insect | Landmark | Plant | Sport | Overall |
|---|---|---|---|---|---|---|---|---|---|
| ■ SD | 68.8 | 75.0 | 62.5 | 87.5 | 62.5 | 65.6 | 84.4 | 71.9 | 72.2 |
| ■ Imagen | 59.4 | 84.4 | 75.0 | 84.4 | 53.1 | 59.4 | 84.4 | 78.1 | 72.2 |
| ■ Flux | 56.3 | 87.5 | 96.9 | 78.1 | 59.4 | 71.9 | 84.4 | 81.3 | 76.9 |
| ■ Imagen-3 | 78.1 | 87.5 | 93.8 | 84.4 | 71.9 | 71.9 | 84.4 | 96.9 | 83.6 |
| ● Custom-Diff | 59.4 | 75.0 | 65.6 | 78.1 | 71.9 | 65.6 | 81.3 | 59.4 | 69.5 |
| ● DreamBooth | 68.8 | 71.9 | 71.9 | 81.3 | 68.8 | 75.0 | 78.1 | 75.0 | 73.8 |
| ● Instruct-Imagen | 46.9 | 56.3 | 56.3 | 31.3 | 21.9 | 50.0 | 59.4 | 50.0 | 46.5 |

(■: Text-to-Image Models, ●: Retrieval-augmented Models)

## A.3 DETAILS OF THE AUTOMATIC METRIC SCORES

In Figure 4 (Bottom) and Figure 5, we present the automatic metric evaluation for backbone and retrieval-augmented text-to-image models, including image-text alignment and image-entity alignment scores. Here, we include a detailed breakdown of the automatic metrics across eight domains of the KITTEN benchmark in Table 5. Specifically, we include a detailed ablation study of the image-entity alignment metric using cosine similarity scores based on two popular image features: the CLIP-Image and the DINO features.

The retrieval models—Custom-Diff, DreamBooth, and Instruct-Imagen—consistently outperform the backbone models in DINO score across the insect, landmark, and plant domains. In contrast, Custom-Diff shows notably worse DINO and CLIP-T scores in the cuisine and sport domains. These results are consistent with human evaluations.

## A.4 DETAILS OF THE CORRELATION BETWEEN AUTOMATIC AND HUMAN METRICS

To quantify the consistency between automatic metrics and human evaluation results, we computed Pearson and Spearman correlations in Table 2. CLIP-Text and CLIP-Image show moderate alignment with user evaluations. Here, we provide a detailed breakdown of the correlation across eight domains in Table 6. While these metrics show some alignment with human evaluations, discrepancies remain in certain categories. Notably, the correlation between CLIP-T and the instruction-following score in the landmark domain, as well as the correlation between CLIP-I and the faithfulness score in the plant domain, are negative. This underscores the limitations of automatic metrics in fully capturing human judgment. Although DINO demonstrates stronger alignment with user evaluations of faithfulness, correlation variability persists across categories. For instance, correlations

Table 5: Detailed breakdown of automatic metrics.

| Model | Aircraft | Vehicle | Cuisine | Flower | Insect | Landmark | Plant | Sport | Overall |
|---|---|---|---|---|---|---|---|---|---|
| Image-Text Alignment: CLIP-Text | | | | | | | | | |
| ■ SD | 0.341 | 0.354 | 0.343 | 0.336 | 0.322 | 0.330 | 0.332 | 0.342 | 0.338 |
| ■ Imagen | 0.329 | 0.344 | 0.338 | 0.337 | 0.323 | 0.312 | 0.316 | 0.335 | 0.329 |
| ■ Flux | 0.341 | 0.346 | 0.329 | 0.327 | 0.320 | 0.308 | 0.314 | 0.343 | 0.329 |
| ■ Imagen-3 | 0.345 | 0.351 | 0.352 | 0.343 | 0.325 | 0.315 | 0.324 | 0.352 | 0.338 |
| ● Custom-Diff | 0.333 | 0.349 | 0.314 | 0.339 | 0.321 | 0.315 | 0.326 | 0.298 | 0.324 |
| ● DreamBooth | 0.333 | 0.347 | 0.335 | 0.333 | 0.322 | 0.318 | 0.324 | 0.335 | 0.331 |
| ● Instruct-Imagen | 0.293 | 0.324 | 0.315 | 0.320 | 0.295 | 0.294 | 0.302 | 0.316 | 0.307 |
| Image-Entity Alignment: CLIP-Image | | | | | | | | | |
| ■ SD | 0.640 | 0.655 | 0.673 | 0.719 | 0.619 | 0.616 | 0.628 | 0.618 | 0.646 |
| ■ Imagen | 0.678 | 0.650 | 0.671 | 0.733 | 0.630 | 0.632 | 0.630 | 0.545 | 0.646 |
| ■ Flux | 0.652 | 0.653 | 0.663 | 0.738 | 0.609 | 0.614 | 0.618 | 0.561 | 0.639 |
| ■ Imagen-3 | 0.677 | 0.656 | 0.665 | 0.726 | 0.623 | 0.638 | 0.632 | 0.582 | 0.650 |
| ● Custom-Diff | 0.688 | 0.678 | 0.593 | 0.759 | 0.619 | 0.655 | 0.662 | 0.491 | 0.643 |
| ● DreamBooth | 0.698 | 0.676 | 0.692 | 0.742 | 0.660 | 0.681 | 0.657 | 0.589 | 0.674 |
| ● Instruct-Imagen | 0.776 | 0.719 | 0.746 | 0.825 | 0.810 | 0.735 | 0.751 | 0.648 | 0.751 |
| Image-Entity Alignment: DINO | | | | | | | | | |
| ■ SD | 0.449 | 0.540 | 0.367 | 0.386 | 0.181 | 0.306 | 0.189 | 0.379 | 0.350 |
| ■ Imagen | 0.610 | 0.559 | 0.359 | 0.469 | 0.198 | 0.353 | 0.207 | 0.331 | 0.386 |
| ■ Flux | 0.524 | 0.576 | 0.382 | 0.477 | 0.153 | 0.343 | 0.218 | 0.369 | 0.380 |
| ■ Imagen-3 | 0.588 | 0.590 | 0.351 | 0.435 | 0.194 | 0.366 | 0.215 | 0.369 | 0.389 |
| ● Custom-Diff | 0.555 | 0.576 | 0.285 | 0.490 | 0.263 | 0.382 | 0.267 | 0.288 | 0.388 |
| ● DreamBooth | 0.580 | 0.567 | 0.406 | 0.435 | 0.280 | 0.407 | 0.250 | 0.371 | 0.412 |
| ● Instruct-Imagen | 0.758 | 0.711 | 0.539 | 0.646 | 0.526 | 0.545 | 0.450 | 0.482 | 0.582 |

(■: Text-to-Image Models, ●: Retrieval-augmented Models)

are lower in the vehicle and landmark domains, highlighting the need for more accurate automatic metrics to better reflect human evaluations and assess model performance.

Table 6: Per-category correlation between automatic metrics and human evaluation.

| Automatic | Human | Type | Aircraft | Vehicle | Cuisine | Flower | Insect | Landmark | Plant | Average |
|---|---|---|---|---|---|---|---|---|---|---|
| CLIP-T | Instruction | Pearson | 0.138 | 0.499 | 0.105 | 0.788 | 0.567 | -0.299 | 0.564 | 0.337 |
| | | Spearman | 0.168 | 0.454 | 0.286 | 0.836 | 0.554 | -0.166 | 0.558 | 0.384 |
| CLIP-I | Faithfulness | Pearson | 0.330 | 0.195 | 0.100 | 0.215 | 0.528 | 0.356 | -0.051 | 0.239 |
| | | Spearman | 0.548 | 0.323 | 0.238 | 0.287 | 0.503 | 0.623 | -0.143 | 0.340 |
| DINO | Faithfulness | Pearson | 0.655 | 0.367 | 0.549 | 0.551 | 0.277 | 0.680 | 0.492 | 0.510 |
| | | Spearman | 0.575 | 0.311 | 0.735 | 0.430 | 0.210 | 0.700 | 0.565 | 0.504 |

## A.5 DATASET STATISTICS

We present the design of the KITTEN benchmark, which focuses on evaluating faithfulness to knowledge-grounded concepts in Figure 2 and Section 3.2. To ensure diversity, we select entities from eight specialized domains and construct diverse prompts in each domain for evaluation. For each entity, we collect a set of support images as inputs to assess retrieval-augmented models, where support images are used to enhance the model's predictions. In addition, we collect a set of evaluation images for conducting human evaluation. A detailed breakdown of the data statistics is provided in Table 7.

Table 7: Statistics of KITTEN benchmark.

| Domain | #Entities | #Prompts | #Support Images | #Eval Images | # (Entitiy, Prompt) |
|--------|-----------|----------|-----------------|--------------|---------------------|
| Aircraft | 48 | 20 | 469 | 237 | 960 |
| Vehicle | 50 | 20 | 500 | 250 | 1000 |
| Flower | 18 | 20 | 180 | 90 | 360 |
| Insect | 50 | 20 | 500 | 250 | 1000 |
| Plant | 48 | 20 | 480 | 240 | 960 |
| Landmark | 50 | 20 | 500 | 250 | 1000 |
| Cuisine | 31 | 20 | 310 | 155 | 620 |
| Sport | 27 | 20 | 270 | 135 | 540 |

### A.6 HUMAN ANNOTATION INSTRUCTIONS

We provide the complete instructions given to human raters for evaluating the generated images in the KITTEN benchmark.

#### RATER INSTRUCTIONS

In this task, you will be provided with a Prompt, Reference Images, and a Generated Image. Your job is to assess the factual accuracy of the generated image with respect to the prompt and the reference images. The goal is to ensure that the entity described in the prompt is factually correct and accurately represented. While the reference images offer a visual starting point, you may conduct your own research (e.g., Google Search) to clarify the appearance of the entity and ensure its accurate depiction in the generated image.

**Part 1: Reference Alignment**

**Faithfulness to Prompt Entity (Factuality)**

Your first task is to evaluate how faithfully the generated image represents the reference entity. Consider whether the key features and overall appearance of the reference entity are accurately depicted.

*Question:* How faithfully does the generated image represent the entity mentioned in the prompt?

*Candidate Answers:*
1 (Not faithful at all): The generated image does not represent the reference entity at all. There are no discernible visual similarities to the reference entity.
2 (Barely faithful): The generated image faintly represents the reference entity, with significant effort needed to see any resemblance. Minor visual elements may be present, but crucial features or characteristics are missing or significantly misrepresented.
3 (Somewhat faithful): The generated image somewhat represents the reference entity, but it's not prominent. There is a clear visual connection in terms of composition, style, or some key elements, but there are noticeable differences, omissions, or misinterpretations.
4 (Mostly faithful): The generated image mostly represents the reference entity and clearly presents it. The generated image draws strong visual inspiration with a strong connection in terms of overall composition, style, key elements, and/or subject matter, despite some variations in details.
5 (Completely faithful): The generated image fully represents the reference entity accurately. It captures all key elements, composition, and style in a way that is almost identical to the reference entity.

**Open Questions for Reference Alignment**

*Visual Similarities:* Describe any visual similarities between the generated image and the reference images, focusing on elements that enhance the recognizability of the entity. Be specific about shape, color, texture, composition, objects, or overall style.
*Visual Differences:* Describe any differences in the generated image that negatively impact its faithfulness to the entity in the prompt and are not specified by the prompt. Focus on aspects that affect recognizability, and avoid mentioning changes that do not impact identification (e.g., angle or color for cars or aircraft).

**Part 2: Text-Image Adherence**

**Adherence to Prompt Beyond References**

Next, evaluate whether the generated image accurately and comprehensively depicts all aspects of the scene or entity described in the prompt that are not already reflected in the reference images. This involves checking for details in the prompt that go beyond what is shown in the reference images.

*Question:* Does the generated image accurately and comprehensively depict any aspects of the scene or entity described in the prompt that are not already reflected in the reference images?

*Candidate Answers:*

Yes: The generated image accurately and comprehensively depicts aspects of the scene or entity described in the prompt that are not already reflected in the reference images.
No: The generated image fails to accurately and comprehensively depict aspects of the scene or entity described in the prompt that are not already reflected in the reference images.

*If the answer is No:*

*Misalignments:* Explain the misalignments between the generated image and the prompt text. Focus on elements or concepts that are not present in the reference images. Be specific about which aspects are missing, inaccurate, or misrepresented.

**Optional: Open-Ended Feedback**

*Question:* Do you have any other comments or observations about the generated image? (optional)

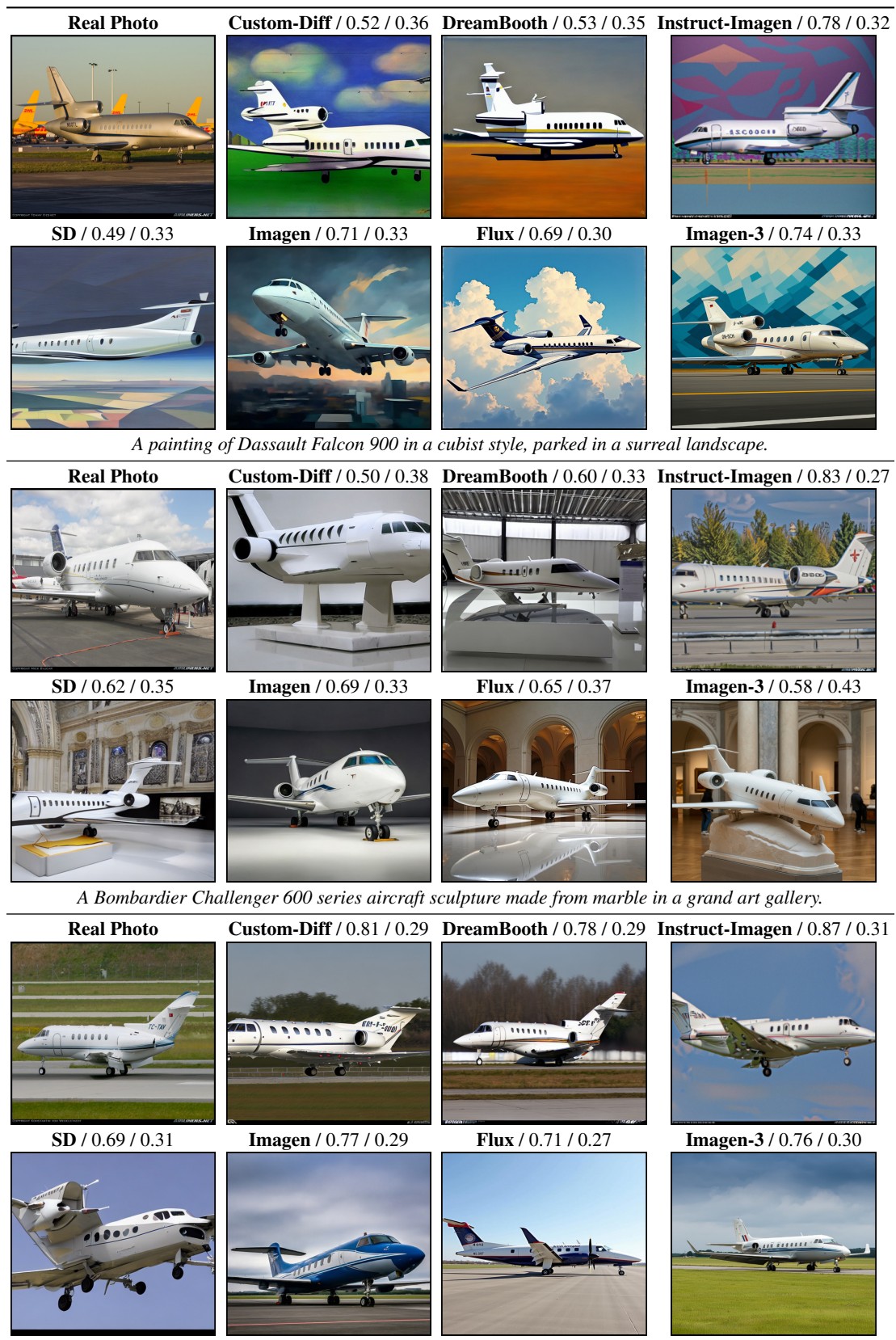

Figure 7: **Qualitative results** for the aircraft domain, including the DINO and CLIP-T scores.

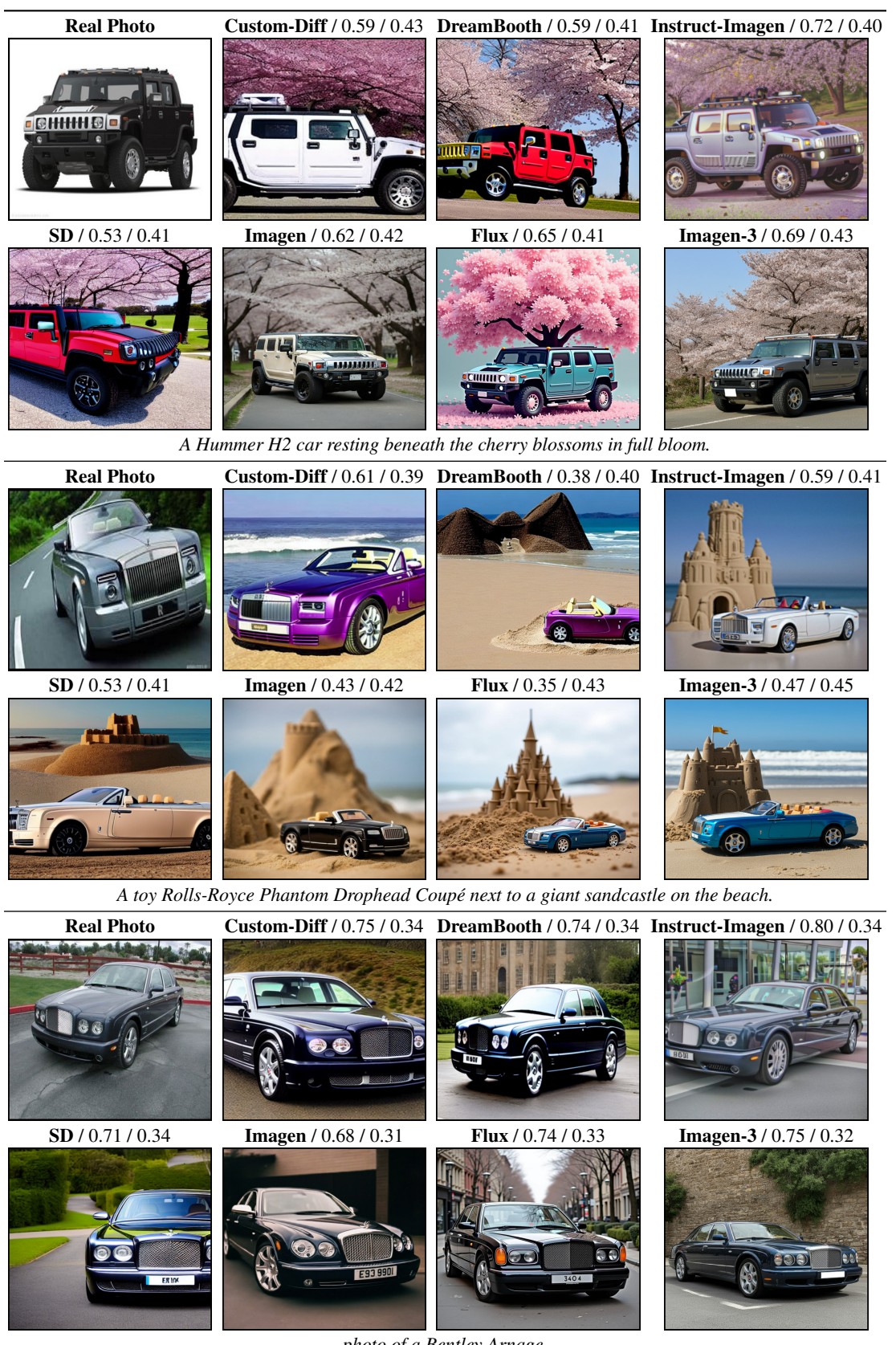

Figure 8: **Qualitative results** for the vehicle domain, including the DINO and CLIP-T scores.

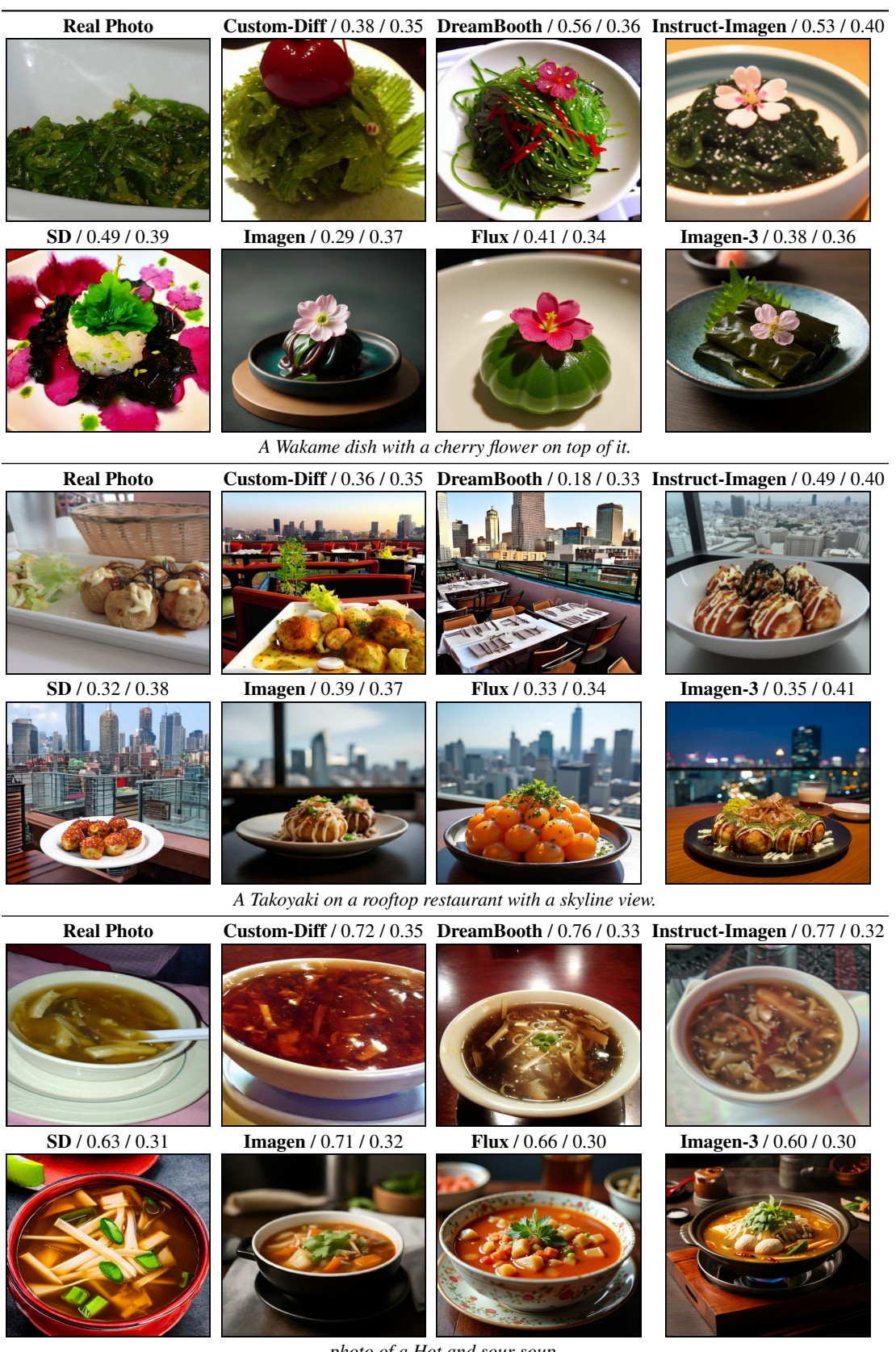

Figure 9: **Qualitative results** for the cuisine domain, including the DINO and CLIP-T scores.

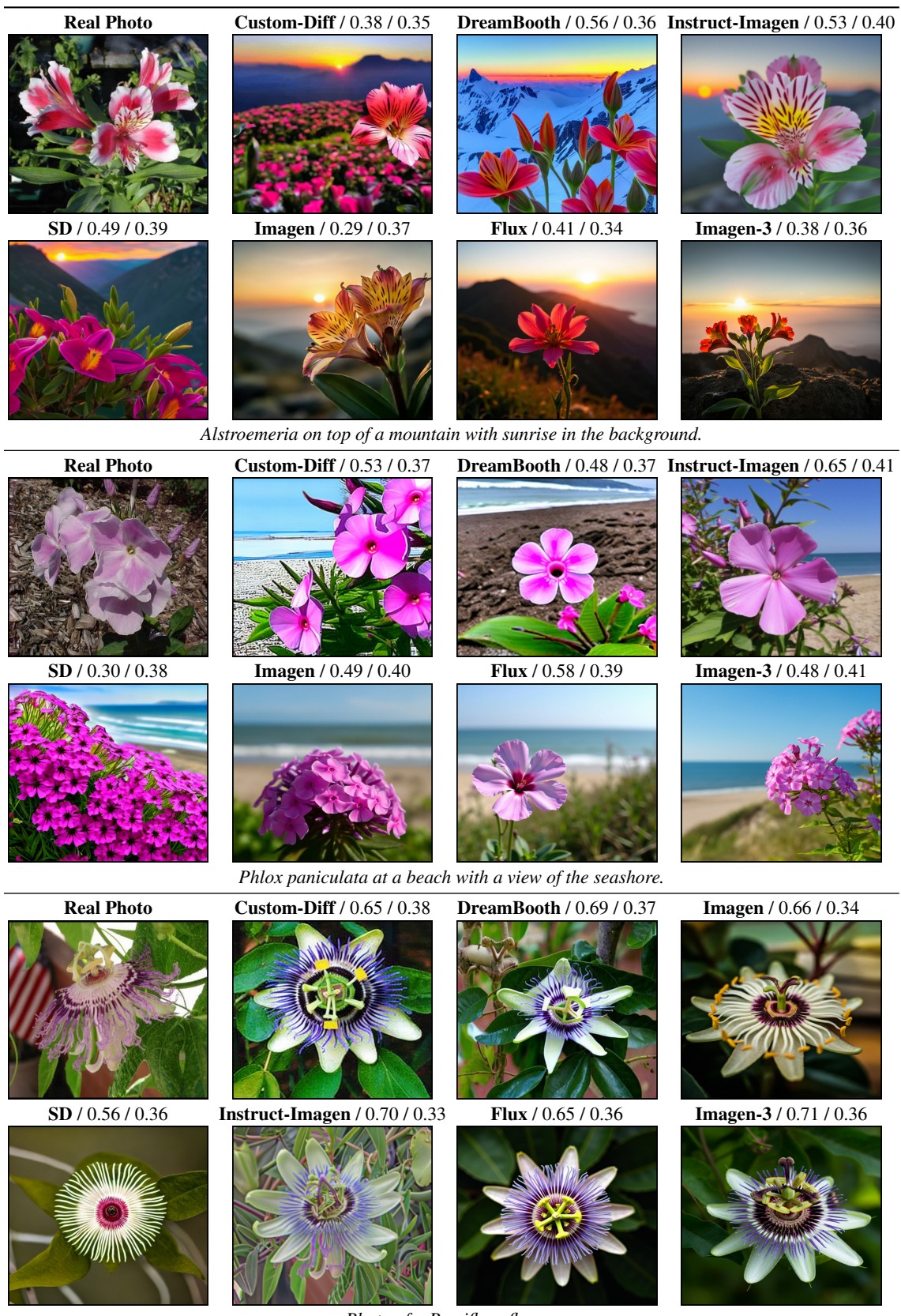

Figure 10: **Qualitative results** for the flower domain, including the DINO and CLIP-T scores.

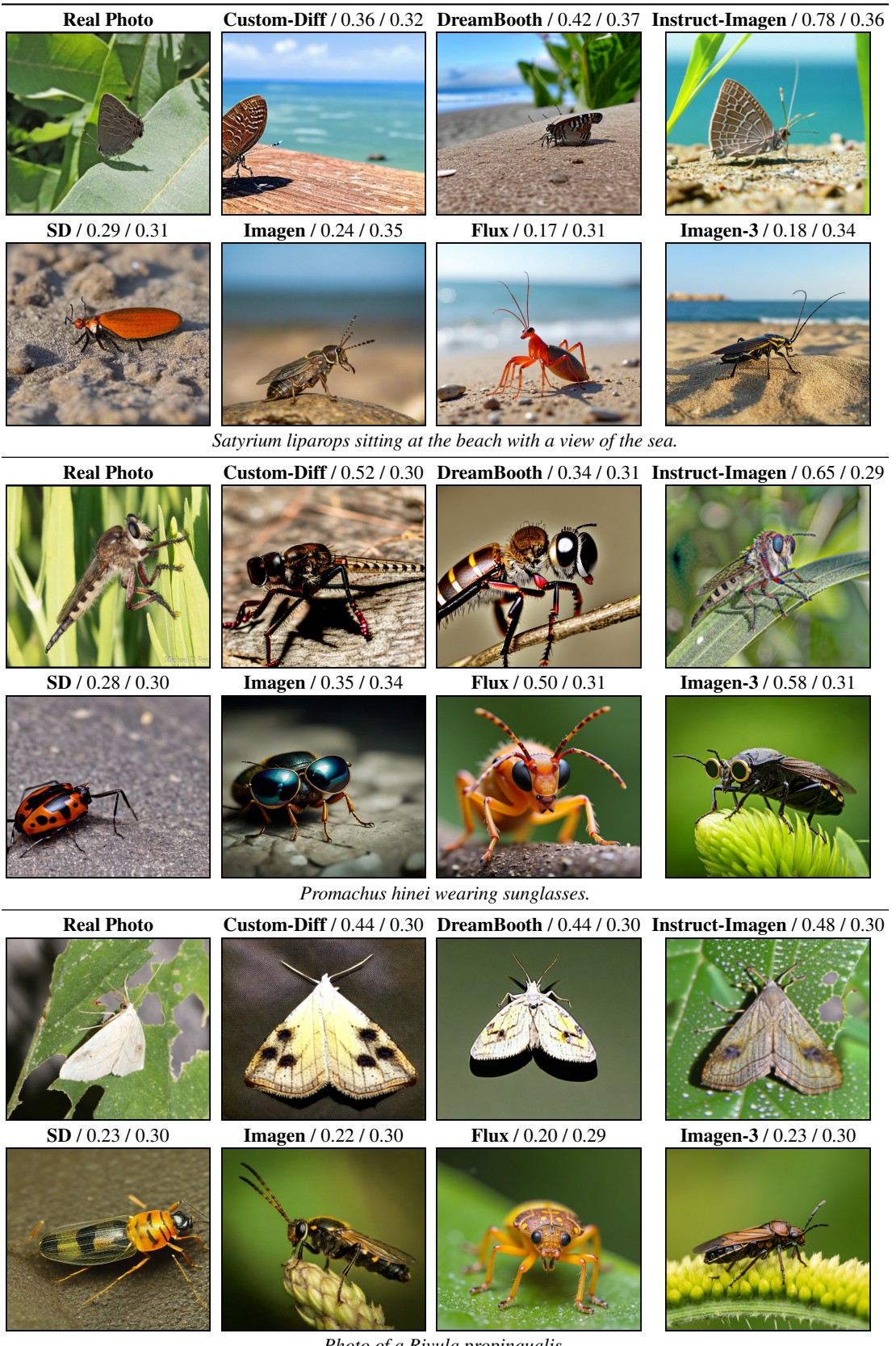

Figure 11: **Qualitative results** for the insect domain, including the DINO and CLIP-T scores.

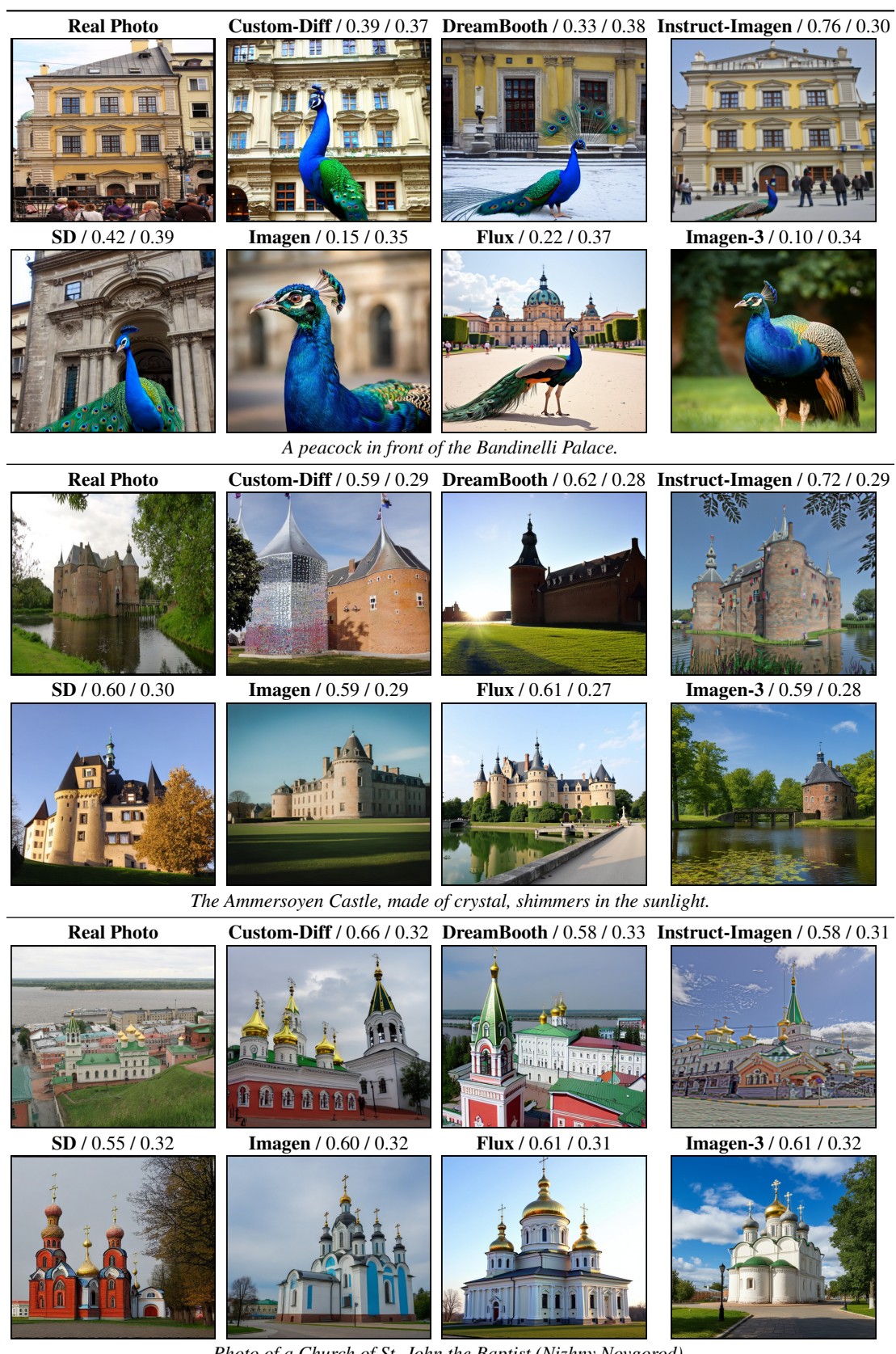

Figure 12: **Qualitative results** for the landmark domain, including the DINO and CLIP-T scores.

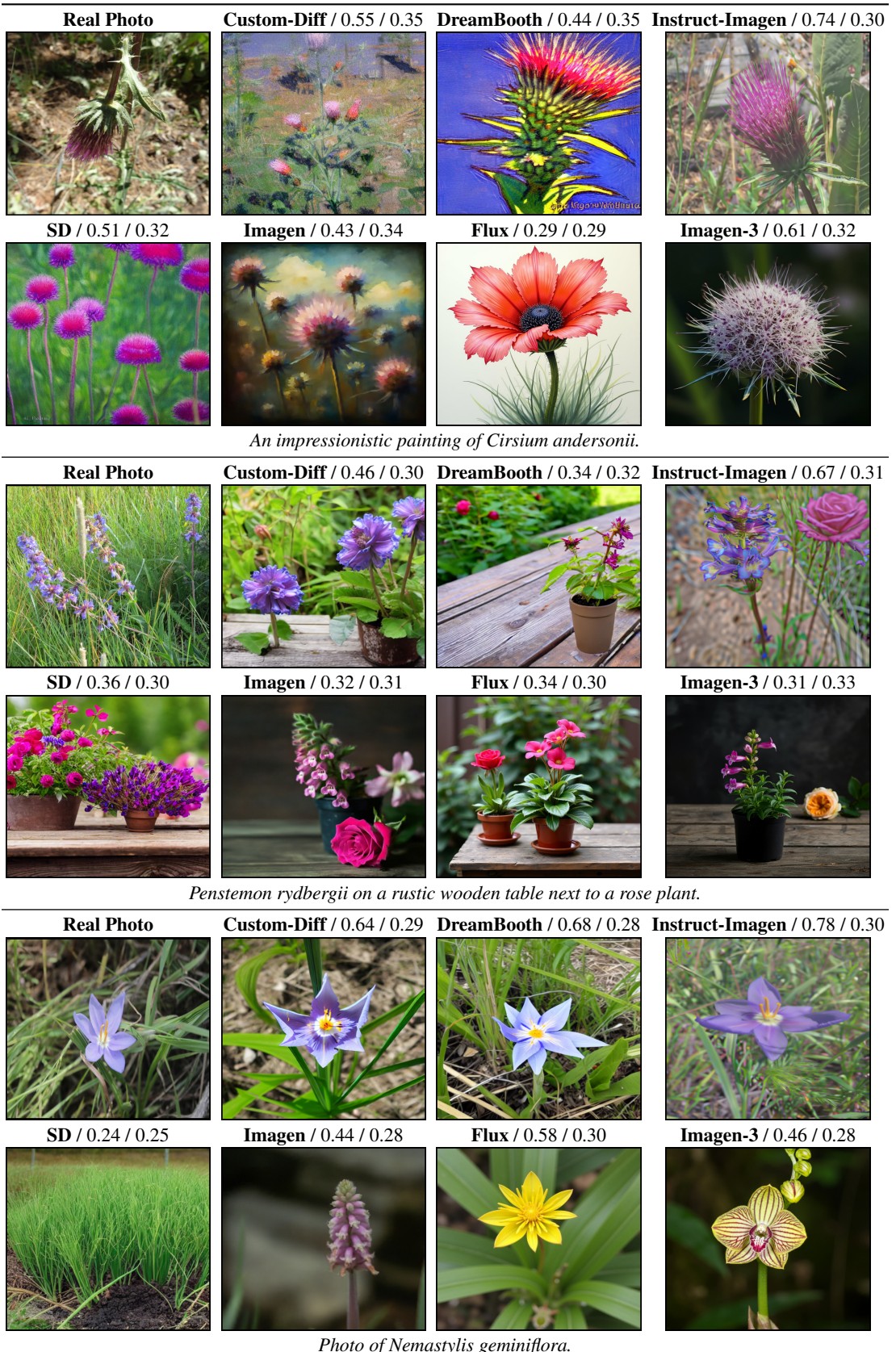

Figure 13: **Qualitative results** for the plant domain, including the DINO and CLIP-T scores.

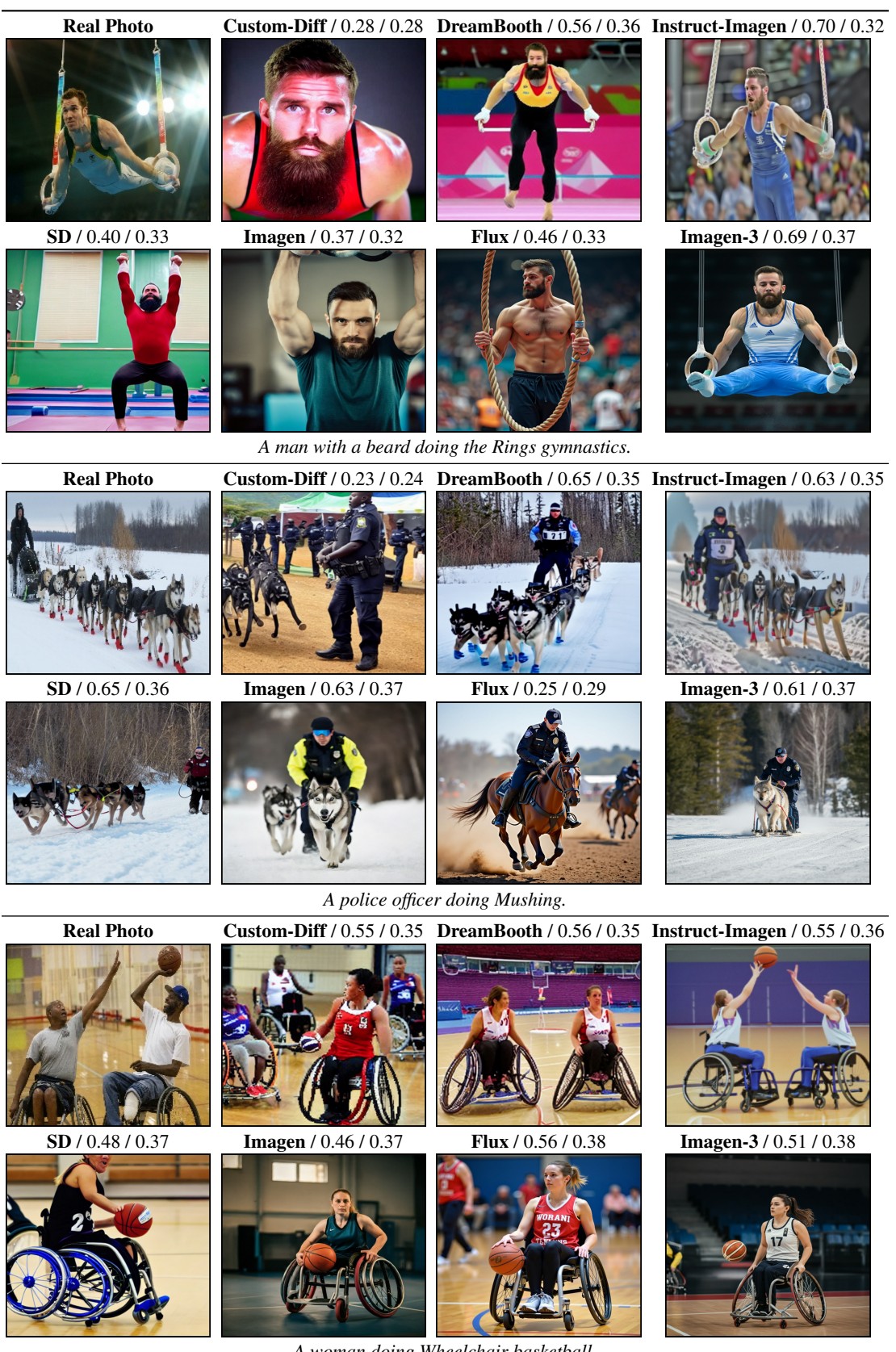

Figure 14: **Qualitative results** for the sport domain, including the DINO and CLIP-T scores.

