# KITTEN 🐱: A KNOWLEDGE-INTENSIVE EVALUATION OF IMAGE GENERATION ON VISUAL ENTITIES SUPPLEMENTARY MATERIALS

## A EXPLORATION OF BALANCING INSTRUCTION-FOLLOWING AND ENTITY FIDELITY

Section 5.1 and Section 5.2 demonstrate that Imagen-3, a strong backbone model, effectively generates specialized entities based solely on text prompts, achieving an instruction-following score of 83.6 and a high faithfulness score of 3.17. Despite these promising results, a gap remains in the fidelity of the generated entities.

To address this, we explore the combination of Imagen-3 with the advanced retrieval-augmented method Instruct-Imagen. This results in a new model, **Instruct-Imagen-3**, which achieves an instruction-following score of 83.21 and an entity fidelity score of 3.48, surpassing Imagen-3's fidelity score of 3.17. Notably, this improvement in entity fidelity is achieved without compromising the model's ability to respond to diverse user inputs. Our results demonstrate that it is possible to balance entity fidelity with creative flexibility, marking a significant step forward in the generation of high-fidelity entities. The results are shown in Figure 1.

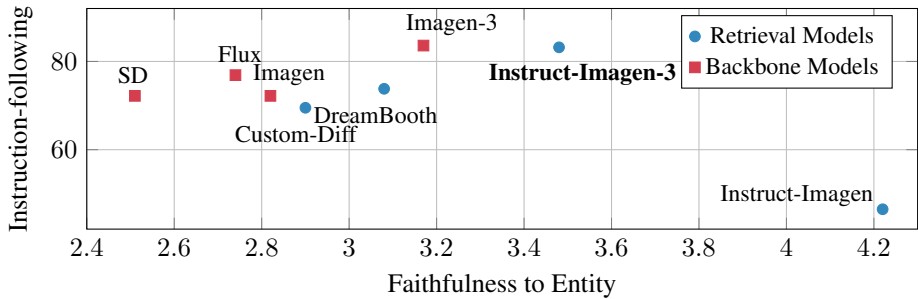

Figure 1: **Human evaluation results** of different text-to-image models. We show the trade-off between Faithfulness to entity and Instruction-following.

Next, we include additional examples of the new model **Instruct-Imagen-3** in Figure 2. In the first example, only Instruct-Imagen and the new model Instruct-Imagen-3 generate a castle that resembles the target entity. Additionally, Instruct-Imagen-3 successfully captures the aspect of "made of crystal," which the other models fail to generate. In the second example, both Instruct-Imagen and Instruct-Imagen-3 generate the target flower and rose, with the new model producing better visual quality. In contrast, other retrieval-augmented approaches, such as Custom-Diff and DreamBooth, fail to generate the rose, while backbone models like SD, Imagen, Flux, and Imagen-3 do not accurately capture the target entity's details. These examples demonstrate that **Instruct-Imagen-3** achieves the coexistence of entity fidelity and creative flexibility.

Our findings highlight future research directions, showing that enhancing the backbone model can improve both instruction-following capability and entity fidelity. Furthermore, combining a strong backbone with an advanced retrieval-augmented method can achieve a coexistence of these two aspects.

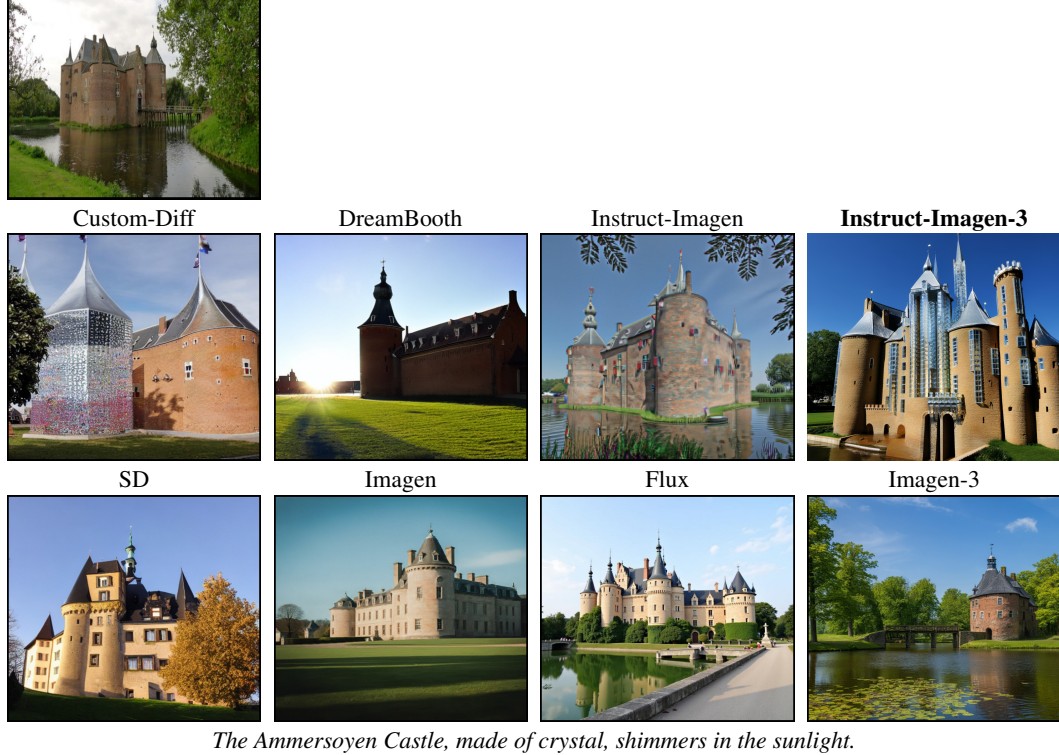

*The Ammersoyen Castle, made of crystal, shimmers in the sunlight.*

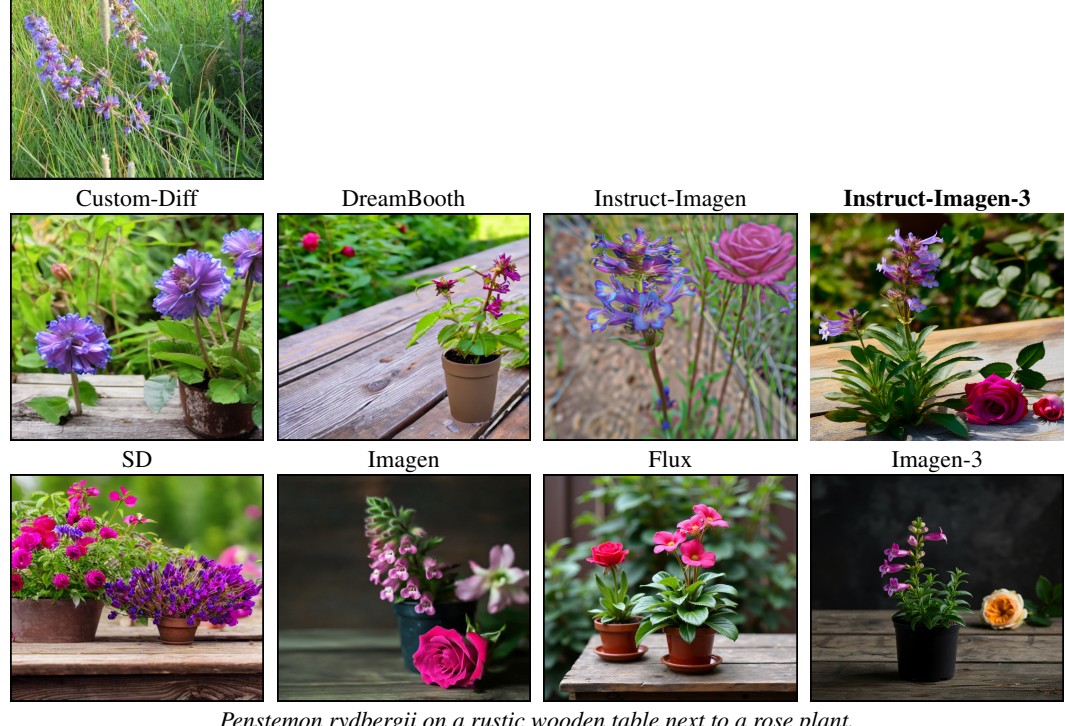

*Penstemon rydbergii on a rustic wooden table next to a rose plant.*

Figure 2: **Qualitative results** for the landmark and the plant domains.