# OpenReview forum: "Kitten: A Knowledge-Intensive Evaluation of Image Generation on Visual Entities"
_ICLR.cc/2025/Conference — ICLR 2025 Conference Withdrawn Submission_

### Official Review · Reviewer_xfyM · 2024-11-02

**Soundness:** 2
**Presentation:** 3
**Contribution:** 2
**Rating:** 5
**Confidence:** 3

**Summary:**

This paper introduces a novel evaluation method to assess image generation models' capability to generate real-world visual entities. The authors propose a benchmark called Kitten and develop both human and automatic evaluation methods to test various image generation models. Through extensive analysis, the author gives a key insight that sota t2i models liake SD, flux exhibit lower faithfulness to the entity, while  retrieval-augmented models show  great advantage.

**Strengths:**

1. The concept of evaluating text-to-image (t2i) models' ability to generate real-world visual entities is intriguing and crucial for progress in this area.
2. The proposed dataset and evaluation criteria offer a valuable resource for the community, enabling more detailed and nuanced evaluations of t2i models.

**Weaknesses:**

1. The paper's novelty is somewhat limited. The insights that retrieval-augmented models show advantage on faithfulness to the entity is understandable, however the finding that they lack instruction-following capabilities is more interesting, have the authors considered exploring methods to improve entity fidelity while maintaining instruction-following capabilities？
2. The dataset covers 8 different domains, Did the authors observe any significant performance variations across different entity domains (e.g., landmarks vs. insects)? If so, what might explain these differences?

**Questions:**

see weaknesses

---

> ### Author Response · Authors · 2024-11-22
>
> **W1. Novelty and contribution.**
>
> Our paper is the first systematic study of a comprehensive set of image-generation models on both entity fidelity and instruction-following. To the best of our knowledge, this evaluation framework is a unique contribution to the field, offering insights that will promote advancements in models that balance knowledge fidelity with instruction-following. Additionally, we provide detailed designs for our human evaluation process to support future research, along with a thorough analysis of current automated metrics to guide the development of more accurate, human-aligned evaluation methods in the future.
>
> Our novel evaluation framework provides insights on model improvement: in Section 5.1 of the paper, these findings facilitate future modeling work to balance instruction-following and entity fidelity: 1) we find that strong backbone models (Imagen-3) can have higher faithfulness than weaker backbone models with retrieval (DreamBooth). 2) DreamBooth is able to improve entity fidelity compared to its baseline model, SD (2.51→3.08), without compromising SD’s instruction-following score (72.2→73.8). However, this is not the case for CustomDiffusion. Our evaluation shows that DreamBooth is a superior retrieval-augmented method compared to CustomDiffusion in terms of balancing fidelity and instruction-following.
>
> ---
> **W1. Exploring the achievability of coexistence and future research directions.**
>
> We explore the combination of a more powerful backbone model, Imagen-3, with the advanced retrieval-augmented method Instruct-Imagen. Preliminary results demonstrate that balancing entity fidelity and creative flexibility is achievable. The new model achieves an instruction-following score of 83.21 and an entity fidelity score of 3.48, surpassing Imagen-3's score of 3.17. Additional examples are included in the supplementary materials.
>
> These findings also highlight future research directions, showing that enhancing the backbone model can improve both instruction-following capability and entity fidelity. Furthermore, combining a strong backbone with a retrieval-augmented method can achieve a coexistence of these two aspects.
>
> ---
>
> **W2. Performance variations across entity domains.**
>
> We show that the performance of each method is domain-dependent in Figure 5 of the paper. We also provide detailed statistics in Appendix A2. In Section 5.2, we provide an analysis of the image-entity alignment scores across domains. For example, retrieval-augmented models generally achieve higher image-entity alignment scores than backbone models in the insect, landmark, and plant domains. This is likely because these domains contain less frequent terms in common image datasets and are therefore underrepresented in the backbone model’s parameters, while retrieval-augmented models improve performance by incorporating reference images during inference.
>
> Additionally, Table 3 shows that the insect and landmark domains have lower average faithfulness to entity scores of 2.12 and 2.61, likely due to the inherent challenges of generating fine-grained details of insects and the many specifications and features of specific landmarks. In contrast, the car category achieves the highest average score of 3.94, possibly due to its higher representation in existing image datasets.

---

> > ### Author Response · Authors · 2024-11-24
> >
> > Dear Reviewer xfyM,
> >
> > Thank you for your comments on our paper. We have provided a response and a supplementary PDF on OpenReview. Since the discussion phase ends on November 26, we would like to know whether we have addressed all the issues. We kindly request you to consider raising the scores after this discussion phase.
> >
> > Thank you!
> >
> > Best,
> > Authors

---

> > > ### Author Response · Authors · 2024-11-25
> > > **Please let us know whether all questions have been answered**
> > >
> > > Dear Reviewer,
> > >
> > > Thank you for your comments on our paper. We have provided a response and a supplementary PDF on OpenReview. Since the discussion phase ends on November 26, we would like to know whether we have addressed all the issues. We kindly request you to consider raising the scores after this discussion phase.
> > >
> > > Thank you!
> > >
> > > Best, Authors

---

> > ### Comment · Reviewer_xfyM · 2024-11-27
> >
> > I appreciate the authors’  detailed explaination, which partially addressed my concerns, however, based on the disscusion from other reviewers, I will keep my original rating.

---

> > > ### Author Response · Authors · 2024-11-27
> > >
> > > Dear Reviewer xfyM,
> > >
> > > We have submitted a new response to Reviewer qEJS and addressed all reviewer questions. We appreciate your feedback on any remaining concerns. We would like to address them and hope this will contribute to a more favorable evaluation.
> > >
> > > Best, Authors

---

> > > > ### Author Response · Authors · 2024-12-02
> > > >
> > > > Dear Reviewer xfyM,
> > > >
> > > > As the discussion period concludes tomorrow (12/02), we kindly remind you to share any remaining concerns regarding our paper. If there are no further issues, we would greatly appreciate it if you could consider updating your score.
> > > >
> > > > We have provided detailed information on the performance variations across entity domains, and we have included results balancing faithfulness to entities and instruction-following in the supplementary PDF, along with detailed performance metrics across prompts in response to Reviewer qEJS.
> > > >
> > > > Our benchmark represents the first systematic study of a comprehensive set of image generation models, focusing on both entity fidelity and instruction-following. Furthermore, we provide a detailed design of our human evaluation process and a thorough analysis of current automated metrics to support future research. We hope these contributions will play a meaningful role for the community in advancing the development of text-to-image generation, particularly in generating diverse visual entities.
> > > >
> > > > Best regards,
> > > > The Authors

---

### Official Review · Reviewer_qEJS · 2024-11-02

**Soundness:** 2
**Presentation:** 3
**Contribution:** 2
**Rating:** 5
**Confidence:** 5

**Summary:**

This paper proposed a novel and significant issue: real-world knowledge in image generation. Focusing on the factual accuracy with which models generate real-world entities, the authors have developed a benchmark based on a diverse set of knowledge entities to systematically evaluate model performance in the domain of real-world knowledge. To more accurately assess the factuality of the images generated by these models, the authors have meticulously designed a set of human evaluation criteria. They conducted a comparative analysis with widely-used automatic metrics, highlighting the deficiencies of these automatic metrics and demonstrating the superior accuracy of human evaluations. The authors evaluated a variety of text-to-image models and retrieval-augmented customization models, uncovering significant deficiencies in the visual representation of world knowledge within these models.

**Strengths:**

1. The paper presents a novel problem in the field of image generation, focusing on the factuality of real-world knowledge in image generation, which is rarely explored in the existing literature.
2. The results are clearly presented, with sufficient tables and visual images to illustrate the findings.
3. The paper is well-written, with a clear and logical organization.

**Weaknesses:**

1. Insufficient Experimental Setup: The current experiments are limited to real-world entities in only 8 domains and 322 knowledge entities, which lacks diversity. This number is significantly less than the variety of entities present in the real world and does not comprehensively reflect the model's performance, potentially leading to data bias. The OntoNotes and WordNet datasets include a more diverse range of categories and entities. If you only consider your 8 domains and limited number of entities, might this affect the generalizability of the results?
2. High Costs: The paper proposes a manual evaluation method, but conducting a single evaluation using this benchmark requires more than 6,000 human assessments, resulting in significant costs.
3. Limited Prompt Types: The paper includes only four types of prompts. It's unclear whether there is a theoretical basis for selecting these four types, and whether high scores in these prompts can fully reflect the model's true understanding of the knowledge entities.
4. Lack of Comparison with Other Benchmarks: The paper does not provide a clear comparison with recent benchmarks in the field, making it difficult to understand the similarities and differences between this benchmark and others (eg. HELM, GenAI-Bench and ConceptMix)

**Questions:**

1.  Is there a theoretical basis for choosing the specific four types of prompts? How were these types determined, and do they align with the objectives of your study?
2. Is there a rationale or empirical evidence supporting this selection as being representative of real-world scenarios?
3. Have the authors discussed the trade-offs between manual and automated evaluation methods in terms of accuracy and cost? Are there any strategies for expanding the evaluation dataset in the future while reducing costs and maintaining quality?
4. Could you please explain in detail how you selected these four types of prompts and the rationale behind your choices? Can you argue for the extent to which these prompt types cover important scenarios involving knowledge entities? Have you considered adding other types of prompts？
5. How does your benchmark compare with existing benchmarks in terms of design goals and evaluation metrics? Are there particular aspects where you believe your benchmark offers distinct advantages?
6. What steps have been taken to mitigate potential biases resulting from the limited diversity of entities and prompts?

---

> ### Author Response · Authors · 2024-11-22
>
> **W1. Q2. Empirical evidence supporting the selection of domains and entities is representative of real-world scenarios.**
>
> Our domain and entity selection is comprehensive and diverse. We note that datasets such as OntoNotes and WordNet do not provide images of entities, making them unsuitable for image generation evaluation. We sample them from OVEN [1], the most widely covering open-domain image recognition dataset (which is compiled from 14 existing image classification datasets). This results in the largest number of visual entities among all existing datasets. By sampling from OVEN [1], our dataset includes 322 entities from 8 domains and 6,440 prompts, offering broader coverage than alternative evaluation benchmarks [2-7] to reduce data bias and improve the generalizability of the results. Existing benchmarks on image customization [2-7] are not focused on generating knowledge entities and often test on a more limited number of subjects, with only a few hundred evaluation prompts. For example, DreamBench [2], DreamBench-v2 [3], CustomConcept101 [4], DreamBench++ [5], MM-Diff [6], and StoryMaker [7] evaluate 30, 30, 101, 150, 25, and 40 subjects, and 750, 220, 2020, 1350, 500 and 800 prompts, respectively. To the best of our knowledge, no current evaluation benchmarks for visual entity generation cover more entities and prompts than ours. We emphasize this work is an initial step in constructing a large dataset to evaluate model performance in the novel problem of visual entity generation, and we aim to expand its scope in future work.
>
> Additionally, existing vision-and-language benchmarks [8-14] on different problems often include far fewer evaluation prompts. For example, Winoground [8], WHOOPS! [9], LLaVA-Bench [10], Visit-Bench [11], ConTextual [12], VibeEval [13], and Visual Riddles [14] include 400, 90, 500, 576, 500, 269, and 400 prompts, respectively. Our benchmark has provided 6,440 prompts—more than ten times the number found in these benchmarks—ensuring comprehensive coverage of real-world scenarios to accurately reflect the models' understanding of knowledge entities.
>
> ---
> **W2. Justification for the cost of human evaluation.**
>
> We emphasize that KITTEN introduces a novel evaluation problem focused on generating visual entities, measuring both fidelity to entities and instruction-following capability. Since there are no established metrics for this task, human evaluation is a critical first step toward gaining a reliable understanding of different model performances. In addition, we explore two widely-used automatic metrics from prior work, and analyze their alignment with human evaluation. This comparison provides valuable insights into the strengths and limitations of automated evaluations.
>
> Our evaluation framework and dataset represent an important contribution to the field, encouraging future research on developing better automatic evaluation methods and improving image-generation models. To support this, we provide detailed descriptions of our human annotation interface and a comprehensive analysis of a variety of models to assist future research efforts. Lastly, we note that our 6k annotations are not exceptionally large compared to other datasets with extensive human annotations. For example, Sherlock [15] and GenAI-Bench [16] include 363K and 40K human annotations, respectively.
>
>
> ---
> **W3. Q1. Theoretical basis and rationale for choosing the four types of prompts.**
>
> While our work focuses on a novel problem, it is not an isolated effort. Instead, it builds upon prior research in customized image generation [2-7], which emphasizes generating specific entities based on a few reference images. Accordingly, the design of our four types of prompts aligns with common practices from prior works in customized image generation [2-7] and is tailored to address the unique challenges of our problem.
>
> For instance, DreamBench [2] employs prompts that (1) generate a subject in various contexts, (2) create artistic renditions of the subject, (3) modify the subject’s properties (e.g., color and material), and (4) combine the subject with accessories. Similarly, CustomConcept101 [4] evaluates prompts that (1) alter background scenes, (2) insert additional objects into a scene, and (3) generate stylistic variations of the main subject. Our prompt design comprehensively incorporates these key scenarios, ensuring that we evaluate aspects of image generation that are both significant to researchers and representative of real-world applications.
>
> We generate prompts by instructing ChatGPT to propose four types of prompts for each domain. These prompts are then manually refined and adjusted by the authors to better align with the characteristics of each domain. To mitigate potential biases, we further extend our prompt sets to include those used in prior benchmarks [2,4].

---

> ### Author Response · Authors · 2024-11-22
>
> **Q3. Trade-offs between manual and automated evaluation methods.**
>
> While manual evaluation ensures high accuracy, automated methods provide a cost-effective alternative, albeit with slightly lower alignment to human perception. In Section 5.2 of the paper, we present a detailed analysis of two popular automatic metrics. Appendix A3 includes the detailed scores. Our analysis in Section 5.2 shows that automatic metrics generally align with human evaluations, though notable discrepancies exist. These insights represent a valuable contribution to the field, fostering future research into refining automatic evaluation metrics.
>
>
> ---
> **Q4. Strategies for expanding the evaluation dataset and incorporating additional types of prompts.**
>
> Since our prompt generation pipeline is semi-automatic and leverages large language models, future benchmarks can easily extend to include other types of prompts with minimal cost while maintaining quality. For instance, the dataset can be expanded by instructing ChatGPT to propose prompts such as (1) generating entities with specific poses, viewpoints, and expressions and (2) generating multiple specific entities. On the other hand, the domains and entities can be expanded by sampling additional classes from the OVEN [1] dataset, as well as from Wikimedia data.
>
> ---
> **Q4. The prompt designs are aligned with the objectives of our study and cover key scenarios involving knowledge entities.**
>
> To align the prompts with our objective of evaluating knowledge-entity generation, we modified the curated prompts by incorporating the entity names directly into the text. This ensures that all prompts explicitly involve the targeted knowledge entities. For example, the template prompt "A peacock in front of the [landmark]" is transformed into specific examples like "A peacock in front of the [Bandinelli Palace landmark]."
> This process resulted in a dataset comprising 322 entities and 6,440 prompts, significantly expanding the dataset's coverage compared to existing benchmarks. For reference, DreamBench [2], DreamBench-v2 [3], CustomConcept101 [4], DreamBench++ [5], MM-Diff [6], and StoryMaker [7] evaluate 30, 30, 101, 150, 25, and 40 subjects, and 750, 220, 2020, 1350, 500 and 800 prompts, respectively.
>
>
> ---
> **W4. Q5. Comparison with existing benchmarks in terms of design goals and our distinct advantages.**
>
> We outline the comparison of KITTEN to existing benchmarks in Section 2 of the paper. Compared to HELM, GenAI-Bench, and ConceptMix, our benchmark evaluates a distinct task. HELM focuses on evaluating language models, while KITTEN is designed for evaluating image generation models. GenAI-Bench and ConceptMix primarily evaluate image generation models on compositional text prompts (e.g., objects with specific colors, shapes, or spatial relationships), such as "A red novel is placed next to a spoon." In contrast, KITTEN focuses on assessing image generation models' ability to generate fine-grained details of specific visual entities by incorporating entity names into the prompts, such as "A peacock in front of Bandinelli Palace." Our benchmark evaluates a novel problem of generating fine-grained visual entities, which is orthogonal to prior benchmarks on image generation. We will include the discussion in the paper to clarify these distinctions.

---

> > ### Comment · Reviewer_qEJS · 2024-11-25
> >
> > Thank you for your comprehensive and detailed responses to the questions raised. Your answers have alleviated some of my concerns. However, I still have a few questions:
> > 1. The introduction of the "Knowledge" in image generation by Kitten is indeed very innovative. However, the method you've employed in constructing prompts seems to align with previous works such as DreamBench and CustomConcept101. I argue that introducing "Knowledge" should encompass more than just visual features. It may be worthwhile to explore the model’s deeper understanding of Knowledge Entities.
> > 2. The four types of prompts you provided each examine different aspects: simply generating the knowledge entity, the knowledge entity in context, the composition of entities, and creating in different styles/materials. However, the paper does not specify the distribution of these four types of prompts, nor does it detail the performance of each model across these aspects. Have the authors observed any differences among the different models with respect to these four aspects? If so, could you provide insights or explanations for these differences? I recommend sharing more details to enhance the clarity and depth of the analysis.

---

> > > ### Author Response · Authors · 2024-11-27
> > >
> > > Thank you for your valuable comments on constructing the knowledge prompts and analyzing performance across different prompts! We provide the answers below.
> > >
> > > ---
> > >
> > > **Q1. Deeper understanding of Knowledge Entities**
> > >
> > > While textual knowledge involves understanding concepts, KITTEN focuses on evaluating text-to-image models, and the knowledge of visual features is indeed our focus. Our work represents a crucial initial step toward enhancing the knowledge fidelity of these models.
> > >
> > > We have considered prompts that include complex forms of knowledge reasoning, such as multi-hop relational understanding. For instance, a prompt like “generate a photo of the tallest building in Manhattan” requires a deeper level of knowledge comprehension. However, this can often be simplified using a language-based preprocessing step, transforming it into the prompt for the target entity, “generate a photo of the Freedom Tower,” rather than directly using it as a prompt for the image generation models. Therefore, this is not the focus of our study, and we leave it as future work.
> > >
> > > ---
> > >
> > > **Q2. Performance differences on different types of prompts.**
> > >
> > > We present the distribution of different types of prompts and their detailed performance across these aspects in the tables below. The types of prompts include: 1) generating the knowledge entity, 2) the knowledge entity in context, 3) the composition of entities, 4) creation in different styles, and 5) creation in different materials.
> > >
> > >
> > > **Distribution of different types of prompts.**
> > >
> > > |                     | Count | Percentage (%) |
> > > |---------------------|-------|----------------|
> > > | 1) Entity Generation | 295   | 4.58%          |
> > > | 2) Entity in Context | 1969  | 30.57%         |
> > > | 3) Composition of Entity | 1467 | 22.78%       |
> > > | 4) Styles of Entity  | 1365  | 21.20%         |
> > > | 5) Materials of Entity | 1344 | 20.87%         |
> > >
> > > ---
> > >
> > > **Image-Entity Alignment (%) across different types of prompts.**
> > >
> > > | Model           | 1) Entity Generation | 2) Entity in Context | 3) Composition of Entity | 4) Styles of Entity | 5) Materials of Entity | Overall|
> > > |-----------------|----------|----------|----------|----------|----------|---|
> > > | Stable-Diffusion| 0.467    | 0.379    | 0.295    | 0.311    | 0.349    | 0.350 |
> > > | Imagen          | 0.472    | 0.409    | 0.333    | 0.376    | 0.385    | 0.386 |
> > > | Flux            | 0.464    | 0.394    | 0.316    | 0.383    | 0.368    | 0.380 |
> > > | Imagen-3        | 0.483    | 0.412    | 0.317    | 0.403    | 0.391    | 0.389 |
> > > | Custom-Diffusion| 0.578    | 0.411    | 0.348    | 0.359    | 0.400    | 0.388 |
> > > | DreamBooth      | 0.597    | 0.440    | 0.376    | 0.371    | 0.417    | 0.412 |
> > > | Instruct-Imagen | 0.626    | 0.573    | 0.566    | 0.593    | 0.606    | 0.582 |
> > > | **Average**     | **0.527**            | **0.431**            | **0.364**                | **0.399**           | **0.417**              | **0.412**|
> > >
> > >
> > > The performance of each model across different prompts generally aligns with the overall ranking. We observe that "Entity in Context" has the highest average Image-Entity score (0.431), followed by "Materials of Entity" (0.417), "Styles of Entity" (0.399), and "Composition of Entity" (0.364). This ranking highlights the challenges of maintaining entity fidelity when prompts involve complex compositions.
> > >
> > > In the categories of "Entity in Context", "Composition of Entity", and "Materials of Entity", retrieval-augmented models often outperform the base models. Among them, Instruct-Imagen achieves the highest score, followed by DreamBooth and Custom Diffusion. Imagen-3 and Flux surpass their respective predecessors, Imagen and SD. However, in an unexpected result for "Composition of Entity", Imagen (0.333) outperforms Imagen-3 (0.317).
> > >
> > > While most categories show higher Image-Entity Alignment on "Materials of Entity" compared to "Styles of Entity", Imagen-3 and Flux surprisingly perform better on styles. Additionally, other models generally obtain higher Image-Entity scores on "Styles of Entity" than on "Composition of Entity", with DreamBooth being an exception. These results reflect the varying strengths of different models in preserving entity fidelity across diverse prompt types.

---

> > > > ### Author Response · Authors · 2024-11-27
> > > >
> > > > **Image-Text Alignment (%) across different types of prompts.**
> > > >
> > > > | Model           | 1) Entity Generation | 2) Entity in Context | 3) Composition of Entity | 4) Styles of Entity | 5) Materials of Entity | Overall |
> > > > |------------------|----------------------|-----------------------|--------------------------|----------------------|------------------------|---|
> > > > | Stable-Diffusion | 0.308  | 0.342  | 0.333  | 0.346  | 0.332  | 0.338 |
> > > > | Imagen           | 0.304  | 0.338  | 0.329  | 0.328  | 0.317  | 0.329 |
> > > > | Flux             | 0.299  | 0.340  | 0.340  | 0.308  | 0.324  | 0.329 |
> > > > | Imagen-3         | 0.303  | 0.347  | 0.345  | 0.322  | 0.333  | 0.338 |
> > > > | Custom-Diffusion | 0.308  | 0.326  | 0.324  | 0.335  | 0.320  | 0.324 |
> > > > | DreamBooth       | 0.308  | 0.336  | 0.327  | 0.341  | 0.319  | 0.331 |
> > > > | Instruct-Imagen  | 0.302  | 0.312  | 0.311  | 0.305  | 0.291  | 0.307 |
> > > > | **Average**      | **0.305**           | **0.334**             | **0.330**                | **0.326**            | **0.319**              | **0.328**
> > > >
> > > >
> > > > We observe that "Entity in Context" achieves the highest average Image-Text Alignment score (0.334), followed by "Composition of Entity" (0.330), "Styles of Entity" (0.326), and "Materials of Entity" (0.319). This pattern highlights that generating entities in context is relatively easier, while accurately changing materials proves to be more challenging.
> > > >
> > > > In the categories of "Entity in Context", "Composition of Entity", and "Materials of Entity", the base models exhibit higher Image-Text Alignment scores, with Imagen-3 achieving the highest score among them. For retrieval-based models, DreamBooth outperforms Custom Diffusion and Instruct-Imagen. Notably, in "Materials of Entity", Imagen scores lower (0.317) than DreamBooth (0.319) and Custom Diffusion (0.320), indicating that Imagen faces more challenges in this category.
> > > >
> > > > ---
> > > >
> > > > **Score differences for "Styles of Entity" prompts.**
> > > >
> > > > We observe that the distribution of scores in "Styles of Entity" differs from other types of prompts. The retrieval-augmented methods (DreamBooth and Custom Diffusion) and the base model Stable Diffusion receive lower Image-Entity Alignment scores, with values of 0.371, 0.359, and 0.311, respectively. This indicates that models based on Stable Diffusion struggle to generate faithful entities when changing their styles. However, Stable Diffusion achieves the highest Text-Entity Alignment score 0.346 in this category, followed by its retrieval-augmented variants, DreamBooth (0.341) and Custom Diffusion (0.335). This suggests that these models excel at generating accurate styles but may compromise entity fidelity.
> > > >
> > > > In contrast, the base models Imagen-3 and Flux achieve higher Image-Entity Alignment scores but perform lower on Image-Text Alignment scores, further demonstrating a trade-off between entity fidelity and creativity in styles.

---

> ### Author Response · Authors · 2024-11-22
>
> **References**
>
> [1] Hexiang Hu, Yi Luan, Yang Chen, Urvashi Khandelwal, Mandar Joshi, Kenton Lee, Kristina Toutanova, and Ming-Wei Chang. Open-domain visual entity recognition: Towards recognizing millions of Wikipedia entities. In ICCV, 2023.
>
> [2] Nataniel Ruiz, Yuanzhen Li, Varun Jampani, Yael Pritch, Michael Rubinstein, and Kfir Aberman. Dreambooth: Fine tuning text-to-image diffusion models for subject-driven generation. In CVPR, 2023.
>
> [3] Wenhu Chen, Hexiang Hu, Yandong Li, Nataniel Ruiz, Xuhui Jia, Ming-Wei Chang, William W. Cohen. Subject-driven Text-to-Image Generation via Apprenticeship Learning. In NeurIPS, 2023.
>
> [4] Nupur Kumari, Bingliang Zhang, Richard Zhang, Eli Shechtman, and Jun-Yan Zhu. Multi-concept customization of text-to-image diffusion. In CVPR, 2023.
>
> [5] Yuang Peng, Yuxin Cui, Haomiao Tang, Zekun Qi, Runpei Dong, Jing Bai, Chunrui Han, Zheng Ge, Xiangyu Zhang, Shu-Tao Xia. DreamBench++: A Human-Aligned Benchmark for Personalized Image Generation. arXiv preprint arXiv:2406.16855, 2024.
>
> [6] Zhichao Wei, Qingkun Su, Long Qin, Weizhi Wang. MM-Diff: High-Fidelity Image Personalization via Multi-Modal Condition Integration. arXiv preprint arXiv:2403.15059, 2024.
>
> [7] Zhengguang Zhou, Jing Li, Huaxia Li ,Nemo Chen, Xu Tang. StoryMaker: Towards consistent characters in text-to-image generation. arXiv preprint arXiv:2409.12576, 2024.
>
>
> [8] Tristan Thrush, Ryan Jiang, Max Bartolo, Amanpreet Singh, Adina Williams, Douwe Kiela, Candace Ross. Winoground: Probing Vision and Language Models for Visio-Linguistic Compositionality. In CVPR, 2022.
>
> [9] Nitzan Bitton-Guetta, Yonatan Bitton, Jack Hessel, Ludwig Schmidt, Yuval Elovici, Gabriel Stanovsky, Roy Schwartz. Breaking Common Sense: WHOOPS! A Vision-and-Language Benchmark of Synthetic and Compositional Images. In ICCV, 2023.
>
> [10] Haotian Liu, Chunyuan Li, Qingyang Wu, Yong Jae Lee. Visual Instruction Tuning. In NeurIPS, 2023
>
> [11] Yonatan Bitton, Hritik Bansal, Jack Hessel, Rulin Shao, Wanrong Zhu, Anas Awadalla, Josh Gardner, Rohan Taori, Ludwig Schmidt. VisIT-Bench: A Benchmark for Vision-Language Instruction Following Inspired by Real-World Use. In NeurIPS, 2023.
>
> [12] Rohan Wadhawan, Hritik Bansal, Kai-Wei Chang, Nanyun Peng. ConTextual: Evaluating Context-Sensitive Text-Rich Visual Reasoning in Large Multimodal Models. In ICML, 2024.
>
> [13] Piotr Padlewski, Max Bain, Matthew Henderson, Zhongkai Zhu, Nishant Relan, Hai Pham, Donovan Ong, Kaloyan Aleksiev, Aitor Ormazabal, Samuel Phua, Ethan Yeo, Eugenie Lamprecht, Qi Liu, Yuqi Wang, Eric Chen, Deyu Fu, Lei Li, Che Zheng, Cyprien de Masson d'Autume, Dani Yogatama, Mikel Artetxe, Yi Tay. Vibe-Eval: A hard evaluation suite for measuring progress of multimodal language models. arXiv preprint arXiv:2405.02287, 2024.
>
> [14] Nitzan Bitton-Guetta, Aviv Slobodkin, Aviya Maimon, Eliya Habba, Royi Rassin, Yonatan Bitton, Idan Szpektor, Amir Globerson, Yuval Elovici. Visual Riddles: a Commonsense and World Knowledge Challenge for Large Vision and Language Models. arXiv preprint arXiv:2407.19474, 2024.
>
> [15] Jack Hessel, Jena D. Hwang, Jae Sung Park, Rowan Zellers, Chandra Bhagavatula, Anna Rohrbach, Kate Saenko, Yejin Choi. The Abduction of Sherlock Holmes: A Dataset for Visual Abductive Reasoning. In ECCV, 2022.
>
> [16] Baiqi Li, Zhiqiu Lin, Deepak Pathak, Jiayao Li, Yixin Fei, Kewen Wu, Tiffany Ling, Xide Xia, Pengchuan Zhang, Graham Neubig, Deva Ramanan. GenAI-Bench: Evaluating and Improving Compositional Text-to-Visual Generation. In CVPR Workshop, 2024.

---

> > ### Author Response · Authors · 2024-11-24
> >
> > Dear Reviewer qEJS,
> >
> > Thank you for your comments on our paper. We have provided a response and a supplementary PDF on OpenReview. Since the discussion phase ends on November 26, we would like to know whether we have addressed all the issues. We kindly request you to consider raising the scores after this discussion phase.
> >
> > Thank you!
> >
> > Best,
> > Authors

---

> ### Author Response · Authors · 2024-11-29
>
> We further provide the detailed human evaluation scores across prompts along with the analysis below.
>
> **Instruction-following score.**
>
> | Model                 | 1) Entity Generation | 2) Entity in Context | 3) Composition of Entity | 4) Styles of Entity | 5) Materials of Entity | Overall |
> |-----------------------|----------------------|----------------------|--------------------------|--------------------|------------------------|---------|
> | Stable-Diffusion      | 100.0               | 80.0                | 64.3                   | 80.4             | 53.1                 | 72.2    |
> | Imagen                | 90.0                | 75.6                | 80.4                   | 66.7             | 59.2                 | 72.2    |
> | Flux                  | 90.0                | 90.0                | 83.9                   | 58.8             | 61.2                 | 76.9    |
> | Imagen-3              | 90.0                | 87.8                | 94.6                   | 76.5             | 69.4                 | 83.6    |
> | Custom-Diffusion      | 100.0               | 72.2                | 66.1                   | 84.3             | 46.9                 | 69.5    |
> | DreamBooth            | 90.0                | 77.8                | 67.9                   | 90.2             | 53.1                 | 73.8    |
> | Instruct-Imagen       | 100.0               | 46.7                | 58.9                   | 47.1             | 20.4                 | 46.5    |
> | Instruct-Imagen-3     | 100.0               | 90.0                | 92.9                   | 84.3             | 55.1                 | 83.2    |
> | **Average**           | **95.0**            | **77.5**            | **76.1**               | **73.5**         | **52.3**             | **72.2**|
>
> The Instruction-Following scores across prompts align with the average performance, with Imagen-3 and Instruct-Imagen-3 emerging as the top performers overall. In particular, Instruct-Imagen-3, Flux, and Imagen-3 excel in "Entity in Context," achieving scores of 90, 90, and 87.8, respectively, while Imagen and other models fall below 80. Similarly, Instruct-Imagen-3 and Imagen-3 demonstrate strong performance in "Composition of Entity," with scores of 92.9 and 94.6. These results highlight the robust instruction-following capabilities of Imagen-3 and its retrieval-augmented variant, Instruct-Imagen-3, in these aspects. In contrast, DreamBooth leads in "Styles of Entity" with the highest score of 90.2, followed by Custom-Diffusion, Instruct-Imagen-3, and Stable-Diffusion, which score 84.3, 84.3, and 80.4, respectively. This demonstrates the strong ability of Stable Diffusion and its retrieval-augmented variants to generate accurate styles. For the "Materials of Entity" category, all models struggle to follow instructions effectively, though Imagen-3 achieves a relatively higher score. These observations align with the Image-Text Alignment metric, underscoring the consistency of the models across evaluation criteria.
>
> ---
>
> **Faithfulness to Entity score.**
>
> | Model                 | 1) Entity Generation | 2) Entity in Context | 3) Composition of Entity | 4) Styles of Entity | 5) Materials of Entity | Overall |
> |-----------------------|----------------------|----------------------|--------------------------|--------------------|------------------------|---------|
> | Stable-Diffusion      | 3.08                | 2.87                | 2.03                   | 2.16             | 2.65                 | 2.51    |
> | Imagen                | 3.96                | 3.16                | 2.38                   | 2.53             | 2.76                 | 2.82    |
> | Flux                  | 3.50                | 3.04                | 2.67                   | 2.32             | 2.54                 | 2.74    |
> | Imagen-3              | 4.08                | 3.55                | 2.78                   | 2.79             | 3.13                 | 3.17    |
> | Custom-Diffusion      | 4.34                | 2.77                | 2.73                   | 3.12             | 2.80                 | 2.90    |
> | DreamBooth            | 4.18                | 3.33                | 2.72                   | 2.95             | 2.94                 | 3.08    |
> | Instruct-Imagen       | 4.90                | 4.18                | 4.16                   | 4.23             | 4.21                 | 4.22    |
> | Instruct-Imagen-3     | 4.20                | 3.74                | 3.24                   | 3.31             | 3.33                 | 3.48    |
> | **Average**           | **4.03**            | **3.33**            | **2.84**               | **2.92**         | **3.05**             | **3.12**|
>
> The ranking across prompts is consistent with the overall ranking. Instruct-Imagen achieves the highest score, followed by Instruct-Imagen-3 and Imagen-3. The retrieval-based methods, DreamBooth and Custom-Diffusion, follow next, while the other base models—Imagen, Flux, Stable-Diffusion—score comparatively lower.

---

> > ### Author Response · Authors · 2024-11-30
> > **Please let us know whether we have addressed your questions**
> >
> > Dear Reviewer,
> >
> > Thank you for your comments on our paper. We have provided a response and a supplementary PDF on OpenReview. Since the discussion phase ends on Dec 2, we would like to know whether we have addressed all the issues. We kindly request you to consider raising the scores after this discussion phase.
> >
> > Thank you!

---

> > > ### Author Response · Authors · 2024-12-02
> > >
> > > Dear Reviewer qEJS,
> > >
> > > We have provided additional responses regarding the construction of knowledge prompts, along with detailed performance metrics across different types of prompts and corresponding analysis. As the discussion period concludes tomorrow (12/02), we kindly remind you to share any remaining concerns regarding our paper. If there are no further issues, we would greatly appreciate it if you could consider updating your score.
> > >
> > > Our benchmark represents the first systematic study of a comprehensive set of image generation models, focusing on both entity fidelity and instruction-following. Additionally, we provide a detailed design of our human evaluation process and an in-depth analysis of current automated metrics to support future research. We hope these contributions will play a meaningful role  for the community in advancing the development of text-to-image generation, particularly in generating diverse visual entities.
> > >
> > > Best regards,
> > > The Authors

---

### Official Review · Reviewer_a7bf · 2024-11-04

**Soundness:** 2
**Presentation:** 3
**Contribution:** 2
**Rating:** 5
**Confidence:** 3

**Summary:**

The paper introduces KITTEN, a benchmark specifically designed to evaluate the accuracy of text-to-image generation models in terms of real-world entities, such as landmarks, plants, and animals. on this benchmark, the authors evaluate multiple existing approaches, including both standard text-to-image models and retrieval-augmented models that leverage reference images.  the findings include 1) that even the most advanced text-to-image models often fail to generate entities with accurate visual details, 2) retrieval-based method often over-rely on these references and struggle to produce novel configurations of the entity as requested in creative text prompts.

**Strengths:**

the focus of the paper is interesting, which is to evaluate the text-to-image model in terms of generating real-world entities with some modifications. the benchmark would also be beneficial to the community. the evaluation is based on two aspects. one is the faithfulness to the entity, and the other is the prompt following as the prompt may contain some changes to the entity. the metric is sound and reasonable. beyond, the authors perform a very comprehensive study on how the existing approaches work on these new benchmarks.

**Weaknesses:**

as the main contribution is the benchmark, it may be better to provide more details on how the benchmark is collected. throughout the paper, i only find the information that the benchmark is collected from 8 categories and Table 7 in supl shows the number of prompts in each. it might be recommended to share more details, e.g. how these categories are collected, why it is these 8 categories, is it diverse enough, how the prompt is collected, who collected the prompts, what the prompt length and variance is.

the paper performs lots of evaluation on the existing text-to-image approaches and the retrieval-based approaches. one important metric is the human evaluation, but it might be better to share more details on the human eval, e.g. how many annotators are there for each task, how the variance is, how reliable the evaluation is.

[minor] in Fig 1, the results are all from proprietary company, e.g. adobe firefly, dalle. but in the experiments, all results are based on other approaches, e.g. flux model. there is no overlap here. it might be better to make it consistent.
[minor] the Flux has multiple versions (https://github.com/black-forest-labs/flux), is it dev or pro?

**Questions:**

see weakness.

---

> ### Author Response · Authors · 2024-11-22
>
> **W1. Details on the creation of the benchmark.**
>
> Our domain and entity selection is comprehensive. We sample them from OVEN [1], the most widely covering open-domain image recognition dataset (which is compiled from 14 existing image classification datasets). This results in the largest number of visual entities among all existing datasets. By sampling from OVEN [1], our dataset includes 322 entities from 8 domains and 6,440 prompts, offering broader coverage than alternative evaluation benchmarks [2-7] to reduce data bias and improve the generalizability of the results. Existing benchmarks on image customization [2-7] are not focused on generating knowledge entities and often test on a more limited number of subjects, with only a few hundred evaluation prompts. For example, DreamBench [2], DreamBench-v2 [3], CustomConcept101 [4], DreamBench++ [5], MM-Diff [6], and StoryMaker [7] evaluate 30, 30, 101, 150, 25, and 40 subjects, and 750, 220, 2020, 1350, 500 and 800 prompts, respectively. To the best of our knowledge, no current evaluation benchmarks for visual entity generation cover more entities and prompts than ours. This is just an initial step in constructing a large dataset to evaluate model performance in the novel problem of visual entity generation, and we aim to expand its scope in future work.
>
> We generate prompts by instructing ChatGPT to propose four types of prompts for each domain, categorized as follows: (1) basic descriptions, (2) entities in specified locations, (3) compositions of entities and other objects, and (4) entities in specific styles or materials. The prompts are then manually refined and adjusted by the authors to better align with the characteristics of each domain. To align the prompts with our objective of evaluating knowledge-entity generation, we modified the curated prompts by incorporating the entity names sampled from each domain directly into the text. This ensures that all prompts explicitly involve the targeted knowledge entities. For example, the template prompt "A peacock in front of the [landmark]" is transformed into specific examples like "A peacock in front of the [Bandinelli Palace landmark]." This process resulted in a dataset comprising 322 entities and 6,440 prompts, significantly expanding the dataset's coverage compared to existing benchmarks.
>
> The prompts range from 4 to 24 words, with an average length of 9.91 words and a standard deviation of 2.86. We will include this detailed information in the final version of the paper.
>
> ---
>
> **W2. Details on human evaluation.**
>
> We thank the reviewer for their valuable feedback regarding our human evaluation. As detailed in Section 3.3 and Figure 3, with full instructions in Appendix A6, we employ five annotators per image to ensure robust assessments. The raters are hired through Prolific.com, a third-party rating service. For the binary task ("Adherence to Prompt Beyond References"), we observe agreement among at least 4 out of 5 annotators in 75% of cases and perfect agreement (5 out of 5) in 44% of cases. For the Likert scale task ("Faithfulness to Reference Entity"), we calculate Krippendorff’s Alpha at 0.60, indicating a good agreement for a subjective task of this complexity. Additionally, we achieve an IoU-like score of 0.53, which penalizes outliers and demonstrates moderate consensus, and an average pairwise Cohen’s Kappa of 0.25, reflecting fair pairwise agreement. The average standard deviation of ratings is 0.74, reflecting moderate variability in annotator judgments. Please let us know if further metrics or details are needed. These metrics collectively demonstrate the reliability of our human evaluation, and we will include these details, along with additional examples and explanations, in the camera-ready version.
>
> ---
>
> **Q1. Version of Flux.**
>
> We use Flux.1-dev and will clarify this in the final version of the paper.

---

> ### Author Response · Authors · 2024-11-22
>
> **References**
>
> [1] Hexiang Hu, Yi Luan, Yang Chen, Urvashi Khandelwal, Mandar Joshi, Kenton Lee, Kristina Toutanova, and Ming-Wei Chang. Open-domain visual entity recognition: Towards recognizing millions of Wikipedia entities. In ICCV, 2023.
>
> [2] Nataniel Ruiz, Yuanzhen Li, Varun Jampani, Yael Pritch, Michael Rubinstein, and Kfir Aberman. Dreambooth: Fine tuning text-to-image diffusion models for subject-driven generation. In CVPR, 2023.
>
> [3] Wenhu Chen, Hexiang Hu, Yandong Li, Nataniel Ruiz, Xuhui Jia, Ming-Wei Chang, William W. Cohen. Subject-driven Text-to-Image Generation via Apprenticeship Learning. In NeurIPS, 2023.
>
> [4] Nupur Kumari, Bingliang Zhang, Richard Zhang, Eli Shechtman, and Jun-Yan Zhu. Multi-concept customization of text-to-image diffusion. In CVPR, 2023.
>
> [5] Yuang Peng, Yuxin Cui, Haomiao Tang, Zekun Qi, Runpei Dong, Jing Bai, Chunrui Han, Zheng Ge, Xiangyu Zhang, Shu-Tao Xia. DreamBench++: A Human-Aligned Benchmark for Personalized Image Generation. arXiv preprint arXiv:2406.16855, 2024.
>
> [6] Zhichao Wei, Qingkun Su, Long Qin, Weizhi Wang. MM-Diff: High-Fidelity Image Personalization via Multi-Modal Condition Integration. arXiv preprint arXiv:2403.15059, 2024.
>
> [7] Zhengguang Zhou, Jing Li, Huaxia Li ,Nemo Chen, Xu Tang. StoryMaker: Towards consistent characters in text-to-image generation. arXiv preprint arXiv:2409.12576, 2024.

---

> > ### Author Response · Authors · 2024-11-24
> >
> > Dear Reviewer a7bf,
> >
> > Thank you for your comments on our paper. We have provided a response and a supplementary PDF on OpenReview. Since the discussion phase ends on November 26, we would like to know whether we have addressed all the issues. We kindly request you to consider raising the scores after this discussion phase.
> >
> > Thank you!
> >
> > Best,
> > Authors

---

> > > ### Author Response · Authors · 2024-11-25
> > > **Please let us know whether all questions have been answered**
> > >
> > > Dear Reviewer,
> > >
> > > Thank you for your comments on our paper. We have provided a response and a supplementary PDF on OpenReview. Since the discussion phase ends on November 26, we would like to know whether we have addressed all the issues. We kindly request you to consider raising the scores after this discussion phase.
> > >
> > > Thank you!
> > >
> > > Best, Authors

---

> > > > ### Author Response · Authors · 2024-11-30
> > > > **Please let us know whether there are remaining issues**
> > > >
> > > > Dear Reviewer,
> > > >
> > > > Thank you for your comments on our paper. We have provided a response and a supplementary PDF on OpenReview. Since the discussion phase ends on Dec 2, we would like to know whether we have addressed all the issues. We kindly request you to consider raising the scores after this discussion phase.
> > > >
> > > > Thank you!

---

> > > > > ### Author Response · Authors · 2024-12-02
> > > > >
> > > > > Dear Reviewer a7bf,
> > > > >
> > > > > As the discussion period concludes tomorrow (12/02), we kindly remind you to share any remaining concerns regarding our paper. If there are no further issues, we would greatly appreciate it if you could consider updating your score.
> > > > >
> > > > > We have provided detailed information on the creation of our benchmark and the human evaluation process. Additionally, we have included results balancing faithfulness to entities and instruction-following in the supplementary PDF, along with detailed performance metrics across prompts in response to Reviewer qEJS.
> > > > >
> > > > > Our benchmark represents the first systematic study of a comprehensive set of image generation models, focusing on both entity fidelity and instruction-following. Furthermore, we provide a detailed design of our human evaluation process and a thorough analysis of current automated metrics to support future research. We hope these contributions will play a meaningful role for the community in advancing the development of text-to-image generation, particularly in generating diverse visual entities.
> > > > >
> > > > > Best regards,
> > > > > The Authors

---

### Official Review · Reviewer_9Vtx · 2024-11-04

**Soundness:** 3
**Presentation:** 4
**Contribution:** 3
**Rating:** 8
**Confidence:** 5

**Summary:**

The paper introduces a benchmark dataset, Kitten, designed to evaluate the capability of text-to-image models in generating visuals with precise entity representations across eight distinct visual domains. For each chosen entity, the authors crafted four unique input prompt types, alongside corresponding support and evaluation image sets for each entity.

The paper incorporates human evaluations focusing on entity fidelity, with two primary criteria: (1) the model’s ability to follow instructions accurately, and (2) the faithfulness of generated visuals to the specified entity. Furthermore, the authors employ automatic metrics, including the CLIP-T score and cosine similarity within DINO's feature space, to assess alignment between generated and target images.

State-of-the-art models, such as Flux, Stable Diffusion (SD), Imagen-3, Dreambooth, and Instruct-Imagen, were tested on the dataset. Experimental results highlight the main challengings of backbone models as well as retrieval-augmented customized models. The results suggest future topics, such as enhancing entity fidelity without compromising the instruction-following capabilities of these models.

**Strengths:**

- This paper tackles a unique and underexplored aspect in the text-to-image generation community --evaluating the fidelity of specific entities in generated images. The motivation and evaluation framework are well-grounded.
- Utilizing Wikipedia as a knowledge base is a clever choice, ensuring broad coverage and relevance of entities. The idea makes the entity coverage more scalable for future extension.
- The proposed evaluation approach, incorporating both human and automatic metrics, strengthens the robustness of the study’s findings.
- The paper offers valuable insights and discussions that can guide future work aimed at improving fidelity in text-to-image models.
- The paper is well-written and easy to understand.

**Weaknesses:**

- While the study suggests a direction for future work aimed at balancing entity fidelity with creative flexibility, the notion remains somewhat ambiguous and challenging to envision in practice. Achieving an optimal outcome where both precise entity representation and creative interpretation coexist is complex, and the paper could benefit from a clearer exploration or concrete examples of what such a balance might look like in generated images.
- The paper suggests that the optimal image generation model should balance entity fidelity with creative flexibility, but it remains unclear whether such coexistence is achievable. If it proves difficult or even impossible to represent both precise entity fidelity and creative flexibility simultaneously, this limitation could reduce the paper’s impact. The study would benefit from a deeper investigation into whether these two qualities can genuinely coexist, as this would provide clearer guidance for future research directions and enhance the paper’s practical relevance.

**Questions:**

see weakness

---

> ### Author Response · Authors · 2024-11-22
>
> **W1. Concrete examples of the balance between entity fidelity and instruction-following.**
>
> We have provided examples illustrating a balance between entity fidelity and creative flexibility in the generated images. Specifically, Instruct-Imagen's results for prompts such as "A Wakame dish with a cherry flower on top of it" in Figure 9, "Alstroemeria on top of a mountain with sunrise in the background" in Figure 10, and "Satyrium liparops sitting at the beach with a view of the sea" in Figure 11 demonstrate enhanced entity fidelity while maintaining creative flexibility. These examples are supported by the highest alignment scores across both metrics when compared to other models.
>
> In Section 5.1 and Figure 4, we show that the retrieval model DreamBooth improves entity faithfulness compared to its baseline model, SD (2.51&rarr;3.08), without compromising SD’s instruction-following score (72.2&rarr;73.8). This is another concrete example that indicates it is possible to achieve good entity fidelity without sacrificing creativity.
>
> ---
>
> **W2. Exploring the achievability of coexistence and future research directions.**
>
> We explore the combination of a more powerful backbone model, Imagen-3, with the advanced retrieval-augmented method Instruct-Imagen. Preliminary results demonstrate that balancing entity fidelity and creative flexibility is achievable. The new model achieves an instruction-following score of 83.21 and an entity fidelity score of 3.48, surpassing Imagen-3's score of 3.17. Additional examples are included in the supplementary materials.
>
> These findings also highlight future research directions, showing that enhancing the backbone model can improve both instruction-following capability and entity fidelity. Furthermore, combining a strong backbone with a retrieval-augmented method can achieve a coexistence of these two aspects.

---

> ### Author Response · Authors · 2024-11-25
> **Please let us know whether all the questions have been answered**
>
> Dear Reviewer
>
> Thank you for your comments on our paper. We have provided a response and a supplementary PDF on OpenReview. Please let us know if you have additional concerns or comments, many thanks!
>
> Best,
> Authors

---

> > ### Comment · Reviewer_9Vtx · 2024-11-26
> > **comments**
> >
> > The authors have addressed my initial questions and thus I will keep my rating. However, the authors need to resolve all the other concerns raised by fellow reviewers in order to get in.

---

### Author Response · Authors · 2024-11-22

We thank the reviewers for their valuable comments. We are pleased that the reviewers find our problem interesting (Reviewer a7bf) and underexplored (Reviewers 9Vtx, qEJS), our study comprehensive (Reviewer a7bf), and offering valuable insights (Reviewer 9Vtx). Our benchmark is recognized as valuable to the community (Reviewers a7bf, xfyM), well-grounded, and scalable for future extensions (Reviewer 9Vtx). The paper is also noted as well-written (Reviewers 9Vtx, qEJS).

We encourage all reviewers to refer to our supplementary PDF, which includes our exploration of balancing instruction-following and entity fidelity, as well as potential future research directions.

---

### Note · Authors · 2025-02-15

I have read and agree with the venue's withdrawal policy on behalf of myself and my co-authors.

---

### Meta-Review · Area_Chair_hV2v · 2024-12-20

**Metareview:**

The paper introduces a benchmark to evaluate the generation fidelity of real-world entities (e.g. landmarks, or animals) and presents an analysis comparing state-of-the-art models on this task. The paper was reviewed by four experts in the field. The reviewers acknowledged that the paper is well written and easy to follow (qEJS), its focus is interesting (a7bf, qEJS, xfyM), and its motivation well grounded (9Vtx). Some reviewers appreciated that the benchmark includes both human and automatic metrics (9Vtx) which may help obtain more nuanced evaluations (xfyM). The findings are informative (9Vtx), and the metrics appear sound (a7bf).

The main concerns raised by the reviewers were:
1. Missing concrete examples of the balance between entity fidelity and instruction-following (9Vtx)
2. Missing additional details on how the benchmark is collected and justifications for choices made (a7bf)
3. Missing information related to the human evaluations - e.g. how many annotations per task, variance, reliability of the results (a7bf)
4. Experimental setup appears insufficient, with limited diversity (qEJS), and limited ablations
5. The novelty appear rather limited (xfyM)
5. The proposed evaluation methodology appears highly costly (qEJS)
6. Missing comparisons with existing benchmarks (qEJS)

During rebuttal and discussion, the authors provided examples to illustrate what is requested by the reviewers, shared details on the benchmark creation process and the human evaluations, attempted to justify some of their choices, stratified the performance of the evaluated models into different types of prompts, and argued for the novelty of the paper and its contribution. During discussion, most reviewers were unresponsive despite the numerous intents of the AC to incentivize discussion with authors and among reviewers. Therefore, the AC went over the paper and rebuttal materials in detail, and concluded that the reviewers' concerns were partially addressed. In particular, concerns related to novelty, potential impact of the proposed benchmark, and quality of execution remain unconvincing. For those reasons, the AC leans towards rejection. The AC encourages the authors to consider the feedback received to improve future iterations of their work. Moreover, to fully assess the typical evaluation dimensions of text-to-image models, the authors should consider including a measure of conditional diversity in their analyses.

**Additional Comments On Reviewer Discussion:**

See above.

---

### Decision · Program_Chairs · 2025-01-22

Reject